# GSDME-mediated pyroptosis promotes the progression and associated inflammation of atherosclerosis

Yuanyuan Wei [1], Beidi Lan[1,2], Tao Zheng [1,2], Lin Yang[3], Xiaoxia Zhang[4], Lele Cheng[1,2], Gulinigaer Tuerhongjiang[1,2], Zuyi Yuan [1,2] ✉ & Yue Wu [1,2] ✉

Pyroptosis, a type of Gasdermin-mediated cell death, contributes to an exacerbation of inflammation. To test the hypothesis that GSDME-mediated pyroptosis aggravates the progression of atherosclerosis, we generate *ApoE* and *GSDME* dual deficiency mice. As compared with the control mice, *GSDME⁻/⁻/ApoE⁻/⁻* mice show a reduction of atherosclerotic lesion area and inflammatory response when induced with a high-fat diet. Human atherosclerosis single-cell transcriptome analysis demonstrates that *GSDME* is mainly expressed in macrophages. In vitro, oxidized low-density lipoprotein (ox-LDL) induces GSDME expression and pyroptosis in macrophages. Mechanistically, ablation of *GSDME* in macrophages represses ox-LDL-induced inflammation and macrophage pyroptosis. Moreover, the signal transducer and activator of transcription 3 (STAT3) directly correlates with and positively regulates GSDME expression. This study explores the transcriptional mechanisms of *GSDME* during atherosclerosis development and indicates that GSDME-mediated pyroptosis in the progression of atherosclerosis could be a potential therapeutic approach for atherosclerosis.

Atherosclerosis, a progressive inflammatory disease characterized by lipid accumulation and cell death within the large- or medium-sized arteries, could lead to ischemia of the heart, brain, and extremities[1]. During the formation and progression of atherosclerosis, macrophages affect the features of culprit lesions and are key integrators of inflammatory signals[2,3]. Previous studies have shown that necrotic cell death stimulates atherogenesis by inducing inflammation and necrotic core enlargement[4]. Dying macrophages in advanced atherosclerotic lesions release cellular contents, cytokines, and proteinases into the extracellular space. These factors exacerbate local inflammation and promote plaque disruption and acute cardiovascular events. Thus, macrophage death has been considered a determinant of the fate of atherosclerotic lesions[5,6].

Pyroptosis is an inflammatory cell death characterized by membrane pore formation, swelling, and subsequent cell lysis combined with the release of proinflammatory intracellular contents, including interleukin (IL)−1β, IL-18, and monocyte chemoattractant protein-1 (MCP-1)[7]. Mechanistically, the pyroptosis process involves canonical caspase-1 activation upon various infections and immunological challenges through different inflammasomes[8]. Meanwhile, noncanonical caspase-11/4/5 activation also triggers pyroptosis upon recognition of cytosolic lipopolysaccharides[9,10]. Recently, several studies have shown that pyroptosis is closely related to atherosclerosis and contributes to plaque instability aggravation[11–13]. Indeed, pyroptosis in macrophages may promote necrotic core formation and plaque instability in advanced atherosclerotic lesions[14].

[1]Department of Cardiovascular Medicine, The First Affiliated Hospital of Xi'an Jiaotong University, Xi'an, Shaanxi, China. [2]Key Laboratory of Environment and Genes Related to Diseases, Ministry of Education, Xi'an Jiaotong University, Xi'an, Shaanxi, China. [3]Department of Vascular Surgery, The First Affiliated Hospital, Xi'an Jiaotong University, Xi'an, Shaanxi, China. [4]Department of Pharmacy, The First Affiliated Hospital, Xi'an Jiaotong University, Xi'an, Shaanxi, China. ✉e-mail: zuyiyuan@mail.xjtu.edu.cn; yue.wu@xjtu.edu.cn

However, the molecular mechanisms underlying the process are not fully understood.

Gasdermins are a family of pore-forming effector proteins that cause membrane permeabilization and pyroptosis[15]. Of these family members, gasdermin D (GSDMD) and gasdermin E (GSDME) are key proteins involved in the pathogenesis of pyroptosis[16]. GSDMD serves as a final executor of inflammasome activity. Recent studies have shown that GSDMD-mediated pyroptosis is involved in the initiation, progression, and complications of atherosclerosis that involve the endothelial cells, macrophages, and smooth muscle cells. Thus, targeting pyroptosis could be an approach to treating atherosclerosis[17].

*GSDME*, also known as *ICERE1* or *DFNA5*, was initially recognized as a candidate gene for autosomal dominant non-syndromic hearing loss[18]. It was later found to possess sequence and structural similarities to gasdermins[19]. Previous studies have shown that caspase 3 cleaves GSDME after Asp270 to generate a necrotic N-GSDME fragment that targets the plasma membrane to induce pyroptosis directly or indirectly after the cells show apoptotic morphologies[20]. Chemotherapy drugs that activate caspase 3 were subsequently shown to induce pyroptosis in cell lines with high GSDME levels but apoptosis in GSDME-negative cell lines[21]. In addition, GSDME can convert apoptosis into pyroptosis in a cell type-specific manner, depending on its expression level. Thus, it remains unclear how *GSDME* is transcriptionally regulated and what the cellular function of GSDME is in inflammatory diseases.

In this work, we perform single-cell RNA sequencing on human carotid atherosclerotic plaques and show *GSDME* distribution in atherosclerotic cells. We observe that *GSDME* is upregulated in human advanced atherosclerotic lesions and atherosclerotic lesions of apolipoprotein E deficient (*ApoE*$^{-/-}$) mice fed with a high-fat diet. Mechanistically, we investigate *GSDME* transcriptional regulation in macrophages. Taken together, We indentify that GSDME mediates pyroptosis-related inflammation in atherosclerosis and can serve as a potential targets to treat atherosclerosis.

## Results

### Pyroptosis is involved in human and mouse atherosclerosis

To explore the ultrastructure of atherosclerosis, we obtained atherosclerotic plaques from patients who underwent carotid endarterectomy (Supplementary Fig. 1a, b). These carotid plaques were then examined by transmission electron microscopy. Numerous macrophages were dead or fragmented due to pyroptosis rather than apoptosis because of high phagocytic load (Fig. 1a). A dying macrophage in the lesions was identified by membrane thickening and abundant lysosomes filled with incompletely degraded cell remnants with pore formation in the plasma membrane (Fig. 1b). Most

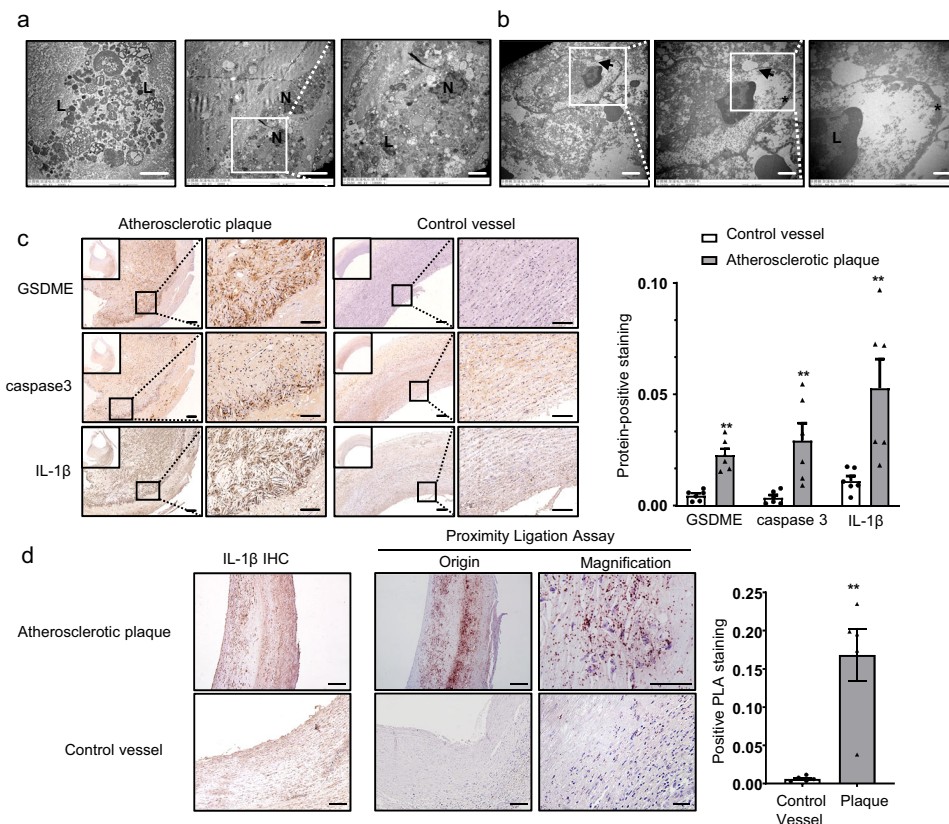

**Fig. 1 | Indications of occurrence of pyroptosis in human and mouse atherosclerotic plaques. a** Electron microscopic appearance of macrophage death in human atherosclerotic plaques. Fragments of a dying macrophage with still recognizable lysosomes (L). Macrophages with normal-looking nucleus (N) and numerous lysosomes (L) containing inclusions of phagocytosed cell remnants. Representative electron microscopic analysis of six different human plaques from one experiment. Scale bar: 5 μm; Scale bar: 2 μm (magnification). **b** Transmission electron microscopy (TEM) images of a suspected pyroptotic macrophage in human carotid artery atherosclerotic plaques. The arrows indicate pore formation and discontinuity of the cell plasma membrane. Scale bar: 2 μm (right), Scale bar: 1 μm (middle), Scale bar: 500 nm (right); Asterisks indicate cell membrane thickening. Representative electron microscopic analysis of six different human plaques from one experiment. **c** Immunohistochemical staining of GSDME, caspase 3, IL-1β in human carotid artery compared with control vessels (human normal abdominal artery derived from autopsy) and the protein integrated optical density (IOD)/area between the two groups are shown (mean ± SEM, *n* = 6). GSDME (\*\**P* = 0.004), caspase 3 (\*\**P* = 0.004), IL-1β (\*\**P* = 0.004). Scale bar: 200 μm; Scale bar: 100 μm (magnification) . **d** In situ detection of the interaction of GSDME and caspase 3 in human carotid atherosclerotic plaques and control vessels (mean ± SEM, *n* = 5) using a generalized proximity ligation assay compared with immunohistochemical staining of IL-1β. \*\**P* = 0.009. For all panels, *P* value was determined by two-tailed Mann–Whitney *U* test. Scale bar: 100 μm; Scale bar: 50 μm (magnification) . Source data are provided as a Source Data file.

macrophages were characterized by ultrastructures typical for cells undergoing pyroptosis with membrane disruption and swelling. Thus, consistent with the previous study[22], although apoptosis takes place in atherosclerotic plaques, the majority of dying cells exhibit membrane disruption and cell lysis (Supplementary Fig. 1c). Meanwhile, we observed lipid cores beneath the fibrous caps and cholesterol clefts in atherosclerotic plaques with inflammatory cells infiltration (Supplementary Fig. 1d). Atherosclerosis also showed increased levels of IL-1β, caspase 3, and GSDME compared with control vessels, as indicated by immunohistochemistry (Fig. 1c). Additionally, we conducted in situ proximity ligation experiments to visualize the interaction between GSDME and caspase 3 in the atherosclerotic plaques and evaluate pyroptosis mediated by GSDME activation and caspase 3. Furthermore, we found similar distribution patterns of overall protein interaction and immunohistochemical IL-1β staining signals (Fig. 1d). This is reminiscent of the relevance of pyroptosis and inflammation. TUNEL-positive nuclei were commonly located in the atherosclerotic lesions of aortic root sections derived from male $ApoE^{-/-}$ mice fed with a high-fat diet for 12 wk (Supplementary Fig. 1e). Taken together, these results strongly suggested that pyroptosis, mediated by GSDME, is involved in atherosclerotic inflammation.

## GSDME is mainly expressed in atherosclerotic macrophages

Heterogeneity and molecular complexity are characteristic features of atherosclerosis. Recent studies have launched the single-cell era, highlighting the diversity of inflammatory cells in plaque progression[23]. To investigate the transcriptome of human atherosclerotic plaques as well as the localization of GSDME in atherosclerosis, we performed single-cell RNA sequence analysis cells from advanced human carotid plaques with calcification or hemorrhage. Standard scRNA-seq filtering excluded cells with a high percentage of mitochondrial genomic transcript reads, indicating potential plasma membrane rupture[24]. Traditionally, this filter is set at 10%. Thus, cells with less than 10% of the mitochondrial genes were retained and 5 370 cells from human carotid artery advanced atherosclerotic lesions of patients undergoing carotid endarterectomy were analyzed. Using uniform manifold approximation and projection (UMAP), we identified 4 clusters of myeloid cells, T lymphocytes, endothelial cells, mast cells, smooth muscle cells, fibroblasts, myofibroblasts, B lymphocytes, and proliferating cells (Fig. 2a). Each cell type showed a distinct gene expression profile (Supplementary Fig. 2a, b). We further analyzed the expression pattern of *GSDME* among the different subgroups in the total atheroma cells. As shown in Fig. 2b, *GSDME* was mainly expressed in M1 macrophage. Furthermore, our study aimed to investigate GSDME-related pyroptosis in atherosclerosis, and some important genes would be ignored if the threshold is too low. *GSDME* expression was positively correlated with mitochondrial gene content indicating GSDME potentiates macrophage pyroptosis (Supplementary Fig. 2c). Therefore, we also chose a threshold of 50% to optimize keeping GSDME expression cells and removing dead and dying cells with reference to several previous studies[25,26]. 6 758 cells were retained in this filter. The result also showed that GSDME was mainly expressed in atherosclerotic macrophages, confirming the validity of the data under the 50% cutoff (Supplementary Fig. 5a–d). Co-staining of GSDME and macrophage marker CD68 (Fig. 2d) also showed that GSDME was localized in CD68-labeled macrophages, consistent with the results of human atherosclerosis single-cell analysis. In addition, GSDME and CD68 expression levels were overtly increased in the atherosclerotic lesion than in the non-lesion aorta. We also investigated co-localizations of GSDME and human macrophage markers in the atherosclerotic aorta and found that GSDME was mainly co-expressed with CD163, CD16, and CD68 in the atherosclerotic areas, and the colocalization coefficients are more than 90% (Supplementary Fig. 1f). These data indicated GSDME is mainly expressed in atherosclerotic macrophages.

Myeloid cells, especially macrophages, are crital in atherosclerotic plaque formation and different subtypes of myeloid cells are developmentally connected[27,28]. To delineate the relationship between different subtypes and investigate genes that regulate the process, we analyzed the pseudo-time developmental of myeloid cells in atherosclerotic plaques. As shown in Fig. 2c, monocytes and dendritic cells were in the early stage of cell development while macrophages were in the late stage. Thus, we inferred that GSDME mainly exerts its function at the late stage in macrophages. Additionally, we performed a pseudo-time developmental analysis of *GSDME* and macrophage marker *CD68* to show the changes during the development of atherosclerosis. The results showed that *CD68* and *GSDME* expression levels were simultaneously increased during the pseudo time (Supplementary Fig. 2d). This finding prompted us to investigate atherosclerotic macrophages. Here, genes differentially expressed in M1 or M2 macrophages were used for GO analysis. M1 and M2 macrophages showed important differences in human atherosclerosis (Supplementary Fig. 3). Next, we examined the biological process involved in M1 and M2 macrophages by gene set enrichment analysis (GSEA). Consistently, we found that genes related to leukocyte migration were upregulated in M1 macrophages, whereas genes related to immune response were downregulated in M2 macrophages (Supplementary Fig. 4a). Furthermore, we also noticed different pathways and biological processes in different subtypes of myeloid cells by performing gene set variation analysis (GSVA) (Supplementary Fig. 4c). As shown in Supplementary Fig. 4b, M1 macrophages expressed several cytokine genes, indicating an inflammatory function. The monocyte cluster and dendritic cells expressed cytokine receptor genes. The abundant expression of cytokine receptor genes allowed monocytes and dendritic cells to receive signals to differentiate into either M1 or M2 macrophages. The dendritic cells and M2 macrophages expressed high levels of major histocompatibility complex genes, indicating the identity of antigen presentation. M1 and M2 macrophages expressed several protease genes, including *MMP19*, *CTSZ*, *CTSL*, and *MMP12*, suggesting participance in tissue remolding. Consistently, analysis with a 50% mitochondrial cut-off shows similar results which confirm the validity of our former data (Supplementary Fig. 5, 6). Taken together, these findings highlight cellular plasticity and the prominence of macrophage GSDME in the progression of human advanced atherosclerosis.

## GSDME expression is increased in human and mouse atherosclerotic plaques

To explore whether GSDME expression was altered in human atherosclerotic plaques, we analyzed 32 human atheroma plaques and nearby macroscopically intact tissues from the gene expression omnibus data sets GSE43292[29] and found that *GSDME* expression was significantly increased in human atheroma plaques (Fig. 3a). To validate this finding, we further examined GSDME expression in human healthy vessels (internal mammary artery) and carotid atherosclerosis plaques. As shown in Fig. 3b, intact GSDME and activated GSDME (N-GSDME) protein levels were significantly increased in human advanced atherosclerotic lesions. Consistently, GSDME expression was also increased in the necrotic area compared with the normal area in the same plaque (Supplementary Fig. 7).

In the atherosclerosis-prone $ApoE^{-/-}$ mice, GSDME protein and mRNA levels were significantly increased after feeding high-fat diet for 12 wk (Fig. 3c, d). Moreover, N-GSDME level was also slightly promoted. Given the close relationship between gasdermins and inflammation[30], we further compared the inflammatory response in the aorta from $ApoE^{-/-}$ mice fed with a high-fat diet or normal diet and wildtype (WT) mice. Consistent with increased *GSDME* mRNA levels, mRNA levels of proinflammatory genes such as *IL-1β*, tumor necrosis factor-α (*TNF*), *MCP-1*, and *IL-6* also significantly increased in the aorta from $ApoE^{-/-}$ mice mentioned above (Fig. 3e). Collectively, these results strongly

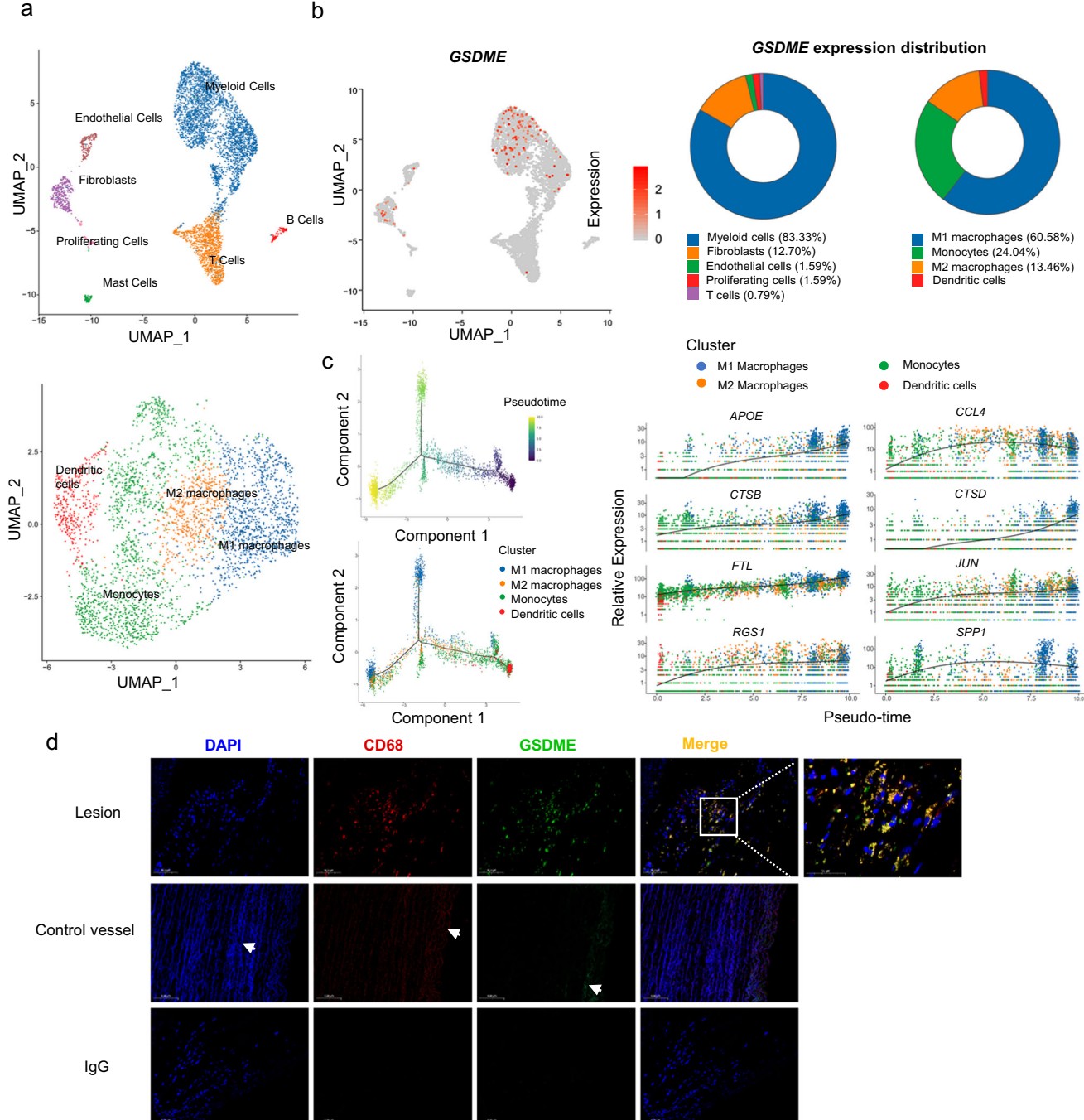

**Fig. 2 | GSDME is mainly expressed in atherosclerotic macrophages.** Cells with over 10% mitochondrial gene content were removed. Single-cell transcriptomic profiling and dissection of the cellular heterogeneity of 5370 cells from human carotid artery advanced atherosclerotic lesions of patients undergoing carotid endarterectomy. **a** Uniform manifold approximation and projection (UMAP) dimensional reduction of the 5370 atheroma cells. Main cell types were identified (upper panel). UMAP distribution of clustering revealed four distinct myeloid populations (lower panel). Population identities were determined based on marker gene expression. **b** Biaxial scatter plots and Pie graphs show the expression pattern of *GSDME* among the different subgroups in the total atheroma cells. Color scale represents expression levels; gray: low, red: high. Atherosclerotic cells and myeloid cells subgroups are labeled by colors (Atherosclerotic cells subgroups: blue, myeloid cells; orange, fibroblasts; green, endothelial cells; red, proliferating cells; purple, T cells; brown, mast cells; pink, B cells. Myeloid cells subgroups: blue, M1 macrophages; green, monocyte; orange, M2 macrophage; red, dendritic cells. **c** The myeloid cell development trajectory visualization in a biaxial scatter plot. Color scale represents the development stage, dark colors indicate early development (left upper panel). Pseudo-time developmental analysis demonstrated a branched single-cell trajectory of myeloid cells beginning with monocytes and dendritic cells ((left lower panel). Myeloid cells are labeled by colors. Two-dimensional plots showing the dynamic expression of myeloid cells marker genes (right panel). Scatter plots for example DEGs of myeloid cells depicting expression level as a function of pseudo time score. Each point represents a single-cell. The color scheme depicts a cluster. **d** Representative immunofluorescence image of CD68 and GSDME co-staining in human carotid artery atheroma and normal aorta, *n* = 4/4 atheroma/normal aorta. The arrows indicate nonspecific staining of elastic fibers. Scale bar: 100 μm; Scale bar: 50 μm (magnification). Source data are provided as a Source Data file.

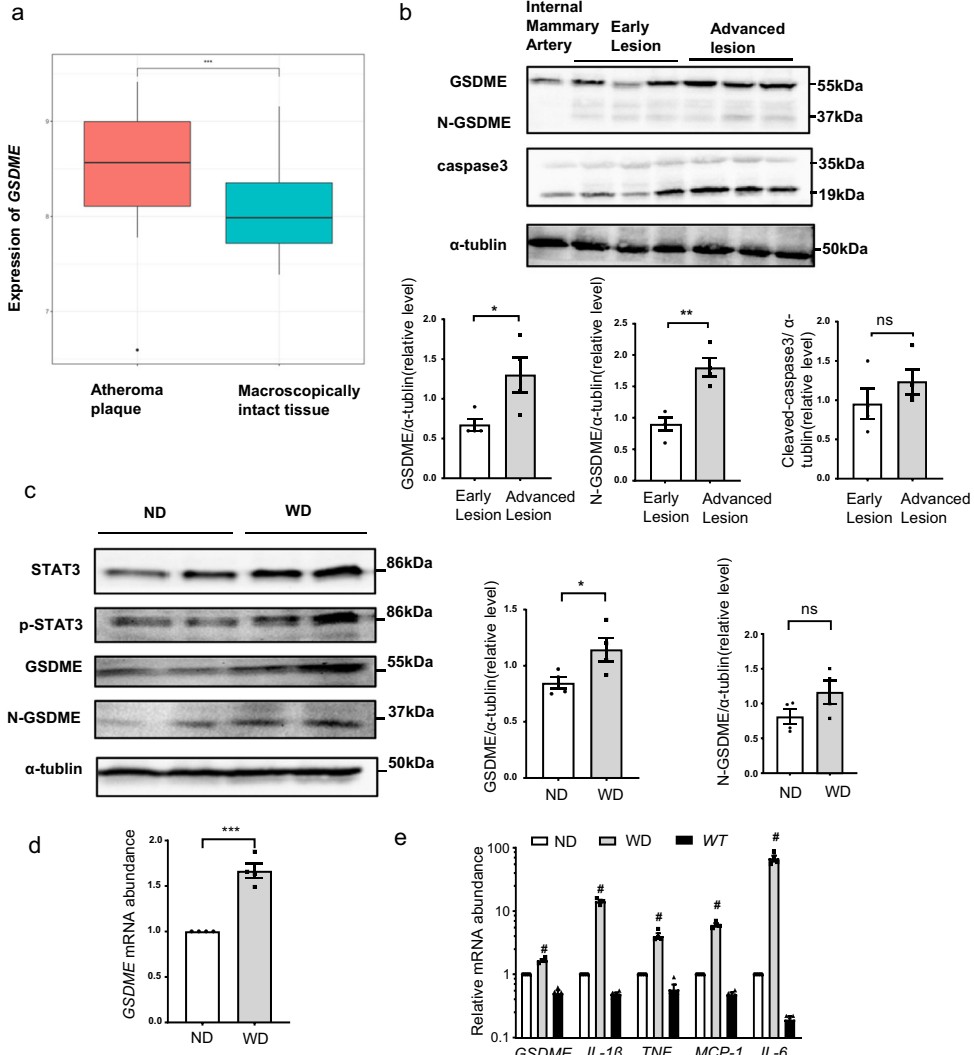

**Fig. 3 | GSDME expression is increased in atherosclerosis. a** *GSDME* gene expression in human carotid atheroma (stage IV) and distant macroscopically intact tissue (stage I and II) (accession no. GSE43292, *n* = 32/group). Boxplots span from the first until the third quartile of the data distribution, and the horizontal line indicates the median value of the data. The whiskers indicate the minimum and maximum values found within 1.5 times the interquartile range (\*\*\**P* < 0.001). **b** Western blotting analysis of GSDME and cleaved caspase 3 expression in human carotid artery atherosclerotic lesions derived from carotid endarterectomy. GSDME (\**P* = 0.036), N-GSDME (\*\**P* = 0.003), cleaved caspase 3 (*P* = 0.303). *n* = 4/group. **c** GSDME protein (\**P* = 0.045), N-GSDME protein (*P* = 0.125), and **d** *GSDME* mRNA (\*\*\**P* < 0.001) were determined by western blotting and quantitative

polymerase chain reaction (qPCR), respectively, in the aortas of male atherosclerosis-prone *ApoE*⁻/⁻ (apolipoprotein E deficient) mice fed with an ND (normal diet) or WD (western diet) for 12 wk. *n* = 4/group from one experiment. ns, not significant. **e** Quantitative polymerase chain reaction (qPCR) analysis of the gene expression relative to inflammation as well as *GSDME* in aortas of *ApoE*⁻/⁻ mice fed with ND or HFD and in control aortas from *WT* mice. *WT*, wild type. #*P* < 0.001 WD *vs* ND or control aortas from *WT* mice. *n* = 4/group. *P* value was determined by unpaired two-tailed Student's *t* test (**a–d**) or one-way ANOVA with Bonferroni post hoc test for multiple comparisons (**e**). Data are expressed as mean ± SEM in **b–e**. Quantification on the blots derive from samples of the same experiment and gels/blots were processed in parallel. Source data are provided as a Source Data file.

suggest that GSDME is crucial in modulating atherosclerotic plaque formation.

### GSDME deficiency attenuates diet-induced atherosclerosis in *ApoE*⁻/⁻ mice

Based on the increased expression of GSDME in atherosclerosis, we wondered whether *GSDME* ablation suppresses the development of atherosclerosis. Hence, 8-wk-old *GSDME* ⁻/⁻/*ApoE*⁻/⁻ mice and control mice (*ApoE*⁻/⁻) were fed on a high-fat diet for 12 wk. As shown in Fig. 4a–c, no significant differences in serum HDL, LDL, and cholesterol levels were observed in these two groups of mice. However, oil red O staining of the entire aorta showed that the atherosclerotic lesion area was reduced by 28% in *GSDME*⁻/⁻/*ApoE*⁻/⁻ mice compared with *ApoE*⁻/⁻ mice (Fig. 4d). Subsequently, sequential 12 aortic sinus cryosections were subjected to hematoxylin-eosin (H&E) and oil red O staining

(Supplementary Fig. 10). Compared with *ApoE*⁻/⁻ mice, a significantly reduced lesion size of both aorta sinus and brachiocephalic artery was observed in *GSDME* ⁻/⁻/*ApoE*⁻/⁻ mice. *GSDME* deficiency also decreased the necrotic lesion area in the brachiocephalic artery (Fig. 4e, f). Further analysis of lesion composition showed that macrophage content assessed by MOMA2 immunostaining was decreased in plaques from *GSDME*⁻/⁻/*ApoE*⁻/⁻ mice than in *ApoE*⁻/⁻ mice. In contrast, Masson trichrome staining of collagen and immunostaining of α-smooth muscle actin (α-SMA) showed no significant difference in collagen and smooth muscle cell content between *GSDME*⁻/⁻/*ApoE*⁻/⁻ mice and *ApoE*⁻/⁻ mice (Supplementary Fig. 8a–e). In addition, the inflammatory responses were alleviated in the aorta from *GSDME*⁻/⁻/*ApoE*⁻/⁻ mice compared with those from *ApoE*⁻/⁻ mice. Furthermore, mRNA levels of proinflammatory genes such as *TNF, IL-1β, IL-6*, and *MCP-1* were significantly reduced in aorta from *GSDME* ⁻/⁻/*ApoE*⁻/⁻ mice (Fig. 4g). Consistently,

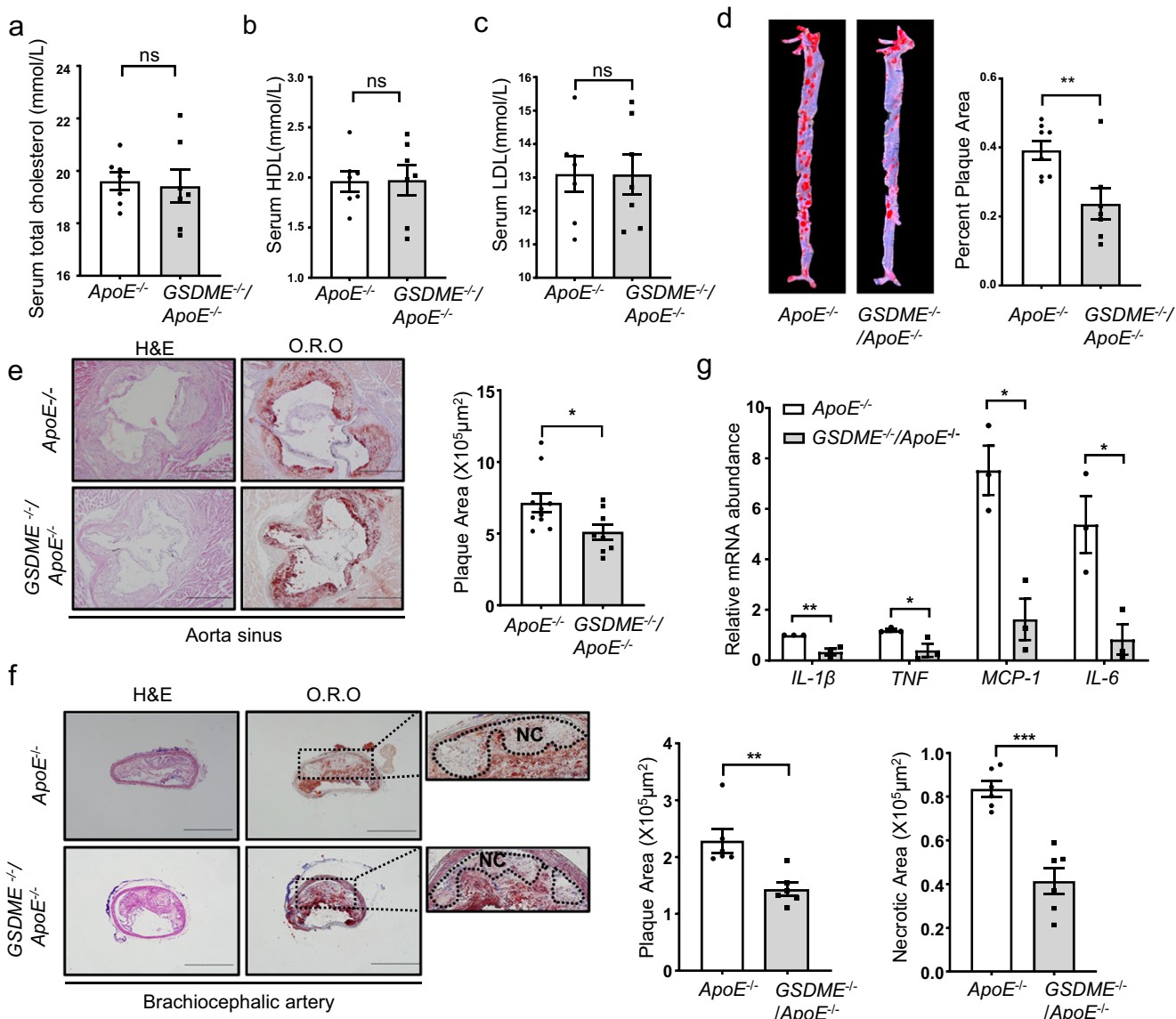

**Fig. 4 | *GSDME* deficiency attenuates atherosclerotic lesion area and size.** Male 8-wk-old *ApoE*[-/-] and *GSDME*[-/-]/*ApoE*[-/-] mice were fed with a high-fat diet for 12 wk. **a** Serum total cholesterol level. *P* = 0.549. **b** Serum total HDL level. *P* = 0.874. **c** Serum total LDL level. *P* = 0.796. (*n* = 7/group). ns, not significant. **d** Representative image of Oil Red O staining of the whole aorta. Plaque area was quantified as the percentage of the total surface area of the aorta, *n* = 8/7 (*ApoE*[-/-] mice / *GSDME*[-/-]/*ApoE*[-/-] mice), **P* = 0.009. **e**, **f** H&E (hematoxylin-eosin) and O.R.O (Oil Red O) staining image of lesions in aortic sinus sections (**P* = 0.031) and brachiocephalic artery (**P* = 0.005, ***P* < 0.001), respectively, and quantitative data

for plaque size and necrotic area, *n* = 10/8 (*ApoE*[-/-] mice /*GSDME*[-/-]/*ApoE*[-/-] mice in **e**), *n* = 6/6 (*ApoE*[-/-] mice / *GSDME*[-/-]/*ApoE*[-/-] mice in **f**). NC, necrotic core. **g** PCR analysis of gene expression related to inflammation in the aortas harvested from male 8-wk-old *ApoE*[-/-] mice and *GSDME*[-/-]/*ApoE*[-/-] mice fed with high-fat diet for 12 wks. *n* = 3/3 (*ApoE*[-/-] mice /*GSDME*[-/-]/*ApoE*[-/-] mice). IL-1β (***P* = 0.008), TNF (**P* = 0.042), MCP-1 (**P* = 0.010), IL-6 (**P* = 0.024). Scale bars are 1 mm (**e**) or 500um (**f**). All panels, data were expressed as mean ± SEM. *P* value was determined by unpaired two-tailed Student's *t* test. Source data are provided as a Source Data file.

serum levels of inflammatory factors, including IL-1β, TNF, and MCP-1, were significantly decreased in *GSDME*[-/-]/*ApoE*[-/-] mice compared to *ApoE*[-/-] mice (Supplementary Fig. 9a–d). Collectively, these results indicate that *GSDME* deficiency mitigates diet-induced atherosclerosis and inflammation in *ApoE*[-/-] mice.

## Ox-LDL induces GSDME expression and pyroptosis in macrophages

To further evaluate the effects of atherosclerotic stimuli on GSDME expression in vitro, primary peritoneal macrophages (PMs) and bone marrow-derived macrophages (BMDMs) were incubated with ox-LDL. As shown in Fig. 5a, GSDME protein level was significantly increased after 24 hours of ox-LDL treatment in PMs. Quantitative real-time PCR analysis showed a similar tendency (Fig. 5b). We further observed the

details of GSDME expression in ox-LDL-treated macrophages and found that ox-LDL promoted GSDME, IL-1β, and IL-18 expression in PMs in a dose-dependent manner (Supplementary Fig. 11a). Consistent with increased GSDME expression, we also observed that ox-LDL promoted a dose-dependent caspase 3/8 expression and activation in macrophages. To further confirm the activation of nucleotide-binding oligomerization segment-like receptor family 3 (NLRP3) inflammasome in ox-LDL-induced pyroptosis, macrophages were pretreated with an NLRP3 inhibitor MCC950. As shown in Supplementary Fig. 11b, pretreatment with MCC950 inhibited ox-LDL-induced NLRP3 and GSDME expression. Consistently, increased GSDME protein and mRNA levels were also detected in ox-LDL-treated BMDMs (Fig. 5c, d). Recent studies demonstrated that GSDME forms pores in the plasma membrane after cleavage by caspase 3,

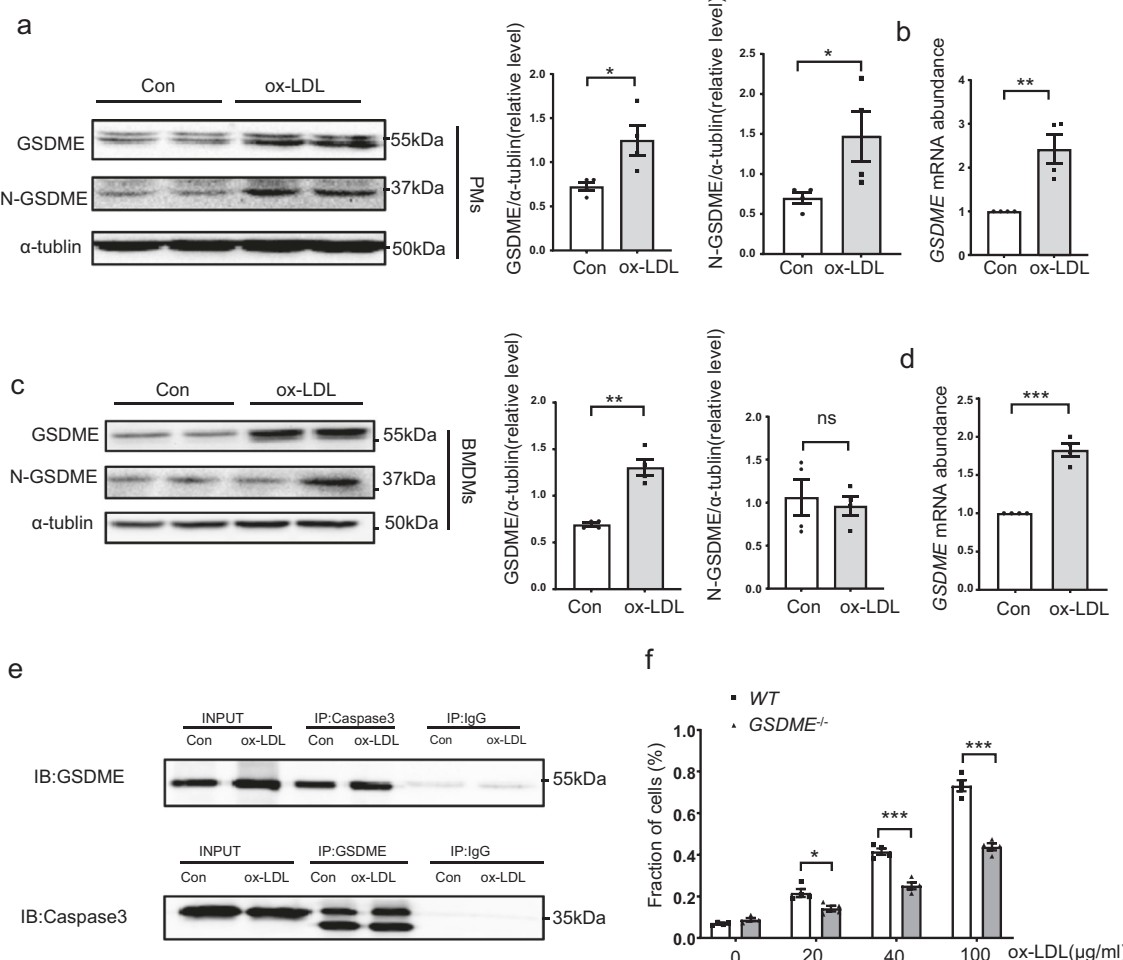

**Fig. 5 | GSDME is induced by ox-LDL in macrophages. a** Western blotting analysis of GSDME and N-GSDME expression in mouse PM treated with ox-LDL (100 μg/ml) for 24 h. GSDME (*P = 0.025), N-GSDME (*P = 0.050), n = 4 independent experiments. **b** Real-time polymerase chain reaction (PCR) analysis of *GSDME* mRNA in level in PMs. **P = 0.005, n = 4 independent experiments. **c** Western blotting analysis of GSDME and N-GSDME expression in BMDMs treated with ox-LDL (100 μg/ml) for 24 h. GSDME (**P = 0.001), N-GSDME (P = 0.688), n = 4 independent experiments. **d** Real-time polymerase chain reaction (PCR) analysis of *GSDME* mRNA in level in BMDM cells. n = 4 independent experiments. ***P < 0.001. **e** GSDME and caspase 3 in whole-cell lysates were pulled down by the appropriate primary antibody and subjected to western blotting analysis to detect GSDME and

caspase 3. Representative for three independent experiments. **f** The LDH content in the culture supernatants of *WT* and *GSDME*[−/−] peritoneal macrophages treated with ox-LDL of indicated concentrations for 24 h. The data shown represent one of four separate experiments (n = 4). *P = 0.018, ***P < 0.001. The fraction of cells was normalized by the experimental LDH release as a percentage of the positive controls. For all panels, unpaired two-tailed Student's *t* test was used for between-group comparisons, and one-way ANOVA with Bonferroni post hoc test for multiple comparisons. Data are expressed as mean ± SEM. Con indicates control. ox-LDL indicates oxidized Low-density lipoprotein. Source data are provided as a Source Data file.

driving a switch from apoptotic to pyroptosis[20,21]. Therefore, we checked the association between caspase 3 and GSDME. As shown in Fig. 5e, the binding of GSDME/caspase 3 was evident in cell lysates pulled down by either caspase 3 or GSDME immunoprecipitation. To assess the extent of ox-LDL-induced macrophage pyroptosis, we measured the activity of lactate dehydrogenase (LDH) released into the culture medium. As shown in Fig. 5f, ox-LDL promoted cell death in a dose-dependent manner. Meanwhile, *GSDME* ablation attenuated LDH release, indicating GSDME was required for ox-LDL-induced pyroptosis. Altogether, our data suggest that GSDME could be upregulated by ox-LDL to induce macrophage pyroptosis and promote the progression of atherosclerosis.

### *GSDME* ablation restrains macrophage migration and inflammation

Based on our findings that *GSDME* ablation suppresses atherosclerosis development in vivo and GSDME binds to caspase 3 to induce pyroptosis in ox-LDL-treated macrophages, we further hypothesized that

*GSDME* ablation might attenuate macrophages' inflammatory behaviors. To test the hypothesis, we treated peritoneal macrophages from *WT* or *GSDME*[−/−] mice with ox-LDL (100 μg/ml) for 24 hours. Cell adhesion-related genes, including *Lama2*, *Pcdhb*, and *Itga7* were significantly downregulated in *GSDME*[−/−] mice as compared with controls (Fig. 6a, b). Consistently, we observed that *GSDME* deficiency downregulated genes related to cell adhesion, which was confirmed by GO analysis (Fig. 6c). Previous studies have demonstrated that cell adhesion to the substrate via a specific adhesion point is essential in cell migration. Thus, we examined whether *GSDME* deficiency alleviates cell migration. Indeed, GSDME deficiency significantly reduced macrophage migration in Transwell (Fig. 6d) and scratch wound assays (Fig. 6e). Given that atherosclerosis-related inflammation is mediated by proinflammatory cytokines, inflammatory signaling pathways and adhesion molecules, we investigated inflammatory response in ox-LDL-treated PMs from *GSDME*[−/−] and wildtype mice. As expected, a significant reduction in the mRNA levels of inflammatory genes such as *TNF*, *IL-1β*, and *MCP-1* was observed in *GSDME*-deficient macrophages

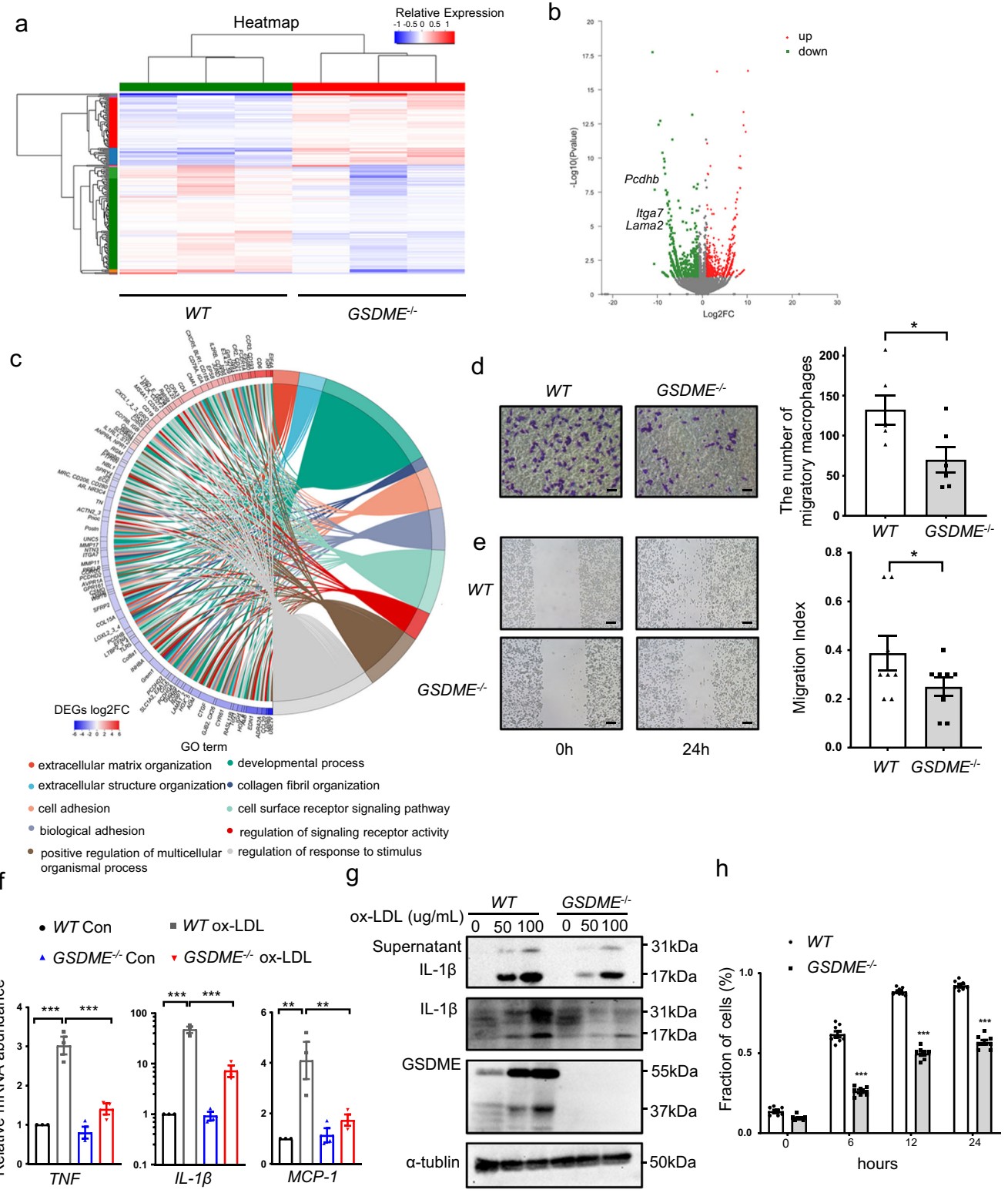

treated with ox-LDL (Fig. 6f). We further explored IL-1β release to the cell-free supernatant by *GSDME⁻/⁻* and wildtype macrophages treated with ox-LDL (Fig. 6g). Immunoblots showed that the level of mature IL-1β released from *GSDME⁻/⁻* macrophages was reduced, consistent with a previous study[31] identifying GSDME as a conduit for IL-1β release. Furthermore, we measured LDH activity in the medium of PMs treated with TNF plus cycloheximide for different periods. As shown in Fig. 6h, *GSDME* deficiency significantly reduced cell death induced by TNF plus cycloheximide, indicating that GSDME determines the switch from apoptosis to pyroptosis. Overall, these results suggest that *GSDME* ablation restrains macrophage inflammatory response and pyroptosis.

## STAT3 upregulates GSDME expression

Multiple studies have demonstrated that STAT3 is essential in the process of atherosclerosis[32]. STAT3 activation is involved in atherosclerotic inflammation, and targeted inhibition of STAT3 can be a potential treatment strategy for atherosclerosis[33]. Raloxifene, a STAT3

**Fig. 6 | Macrophage *GSDME* deficiency represses macrophage inflammation and pyroptosis. a** Transcriptomic analysis of oxLDL-stimulated *WT* and *GSDME*[-/-] macrophages. Peritoneal macrophages were isolated from *WT* and *GSDME*[-/-] mice respectively and treated with oxLDL for 24 hours. Differentially expressed genes (DEGs) were shown by a cluster heat map. Color scale represents relative expression levels; blue: low, red: high. **b** Volcano plots showed the DEGs between *WT* and *GSDME*[-/-] macrophages. Each dot represents a specific gene, with red dots indicating significantly up-regulated genes and green dots indicating significantly downregulated genes. DEGs were identified by a fold change>2 (Log$_2$FC > 1 or Log$_2$FC < −1) and *P* adjust <0.05. **c** GO Chord plot of top 10 ranked overrepresented GO terms. Chords represent the detailed relationship between the expression levels of DEGs (left semicircle parameter) and their enriched GO terms (right semicircle parameter). Color scale represents Log$_2$FC of the DEGs. blue: low, red: high. **d** Representative images (left) of transwell migration assay showing migration of *WT* and *GSDME*[-/-] macrophages with the addition of MCP-1 (600 ng/ml) to the medium in the lower chambers. Bar graph (right) shows the quantitative estimation of the number of migrated cells. *n* = 6/group. *P* = 0.027. Scale bars are 50 µm. **e** Representative images(left) of the wound healing test are shown for *WT* and

*GSDME*[-/-] macrophages. Bar graph (right) shows the migration index. *n* = 8/group. *P* = 0.049. Scale bars are 100 µm. **f** Real-time polymerase reaction (PCR) analysis of gene expression related to inflammation in *WT* and *GSDME*[-/-] mice peritoneal macrophage treated with oxLDL(100 µg/ml) for 24 hours. *n* = 8/group. *TNF* (***P < 0.001), *IL-1β* (***P < 0.001), *MCP-1* (**P = 0.001 *WT* Con *vs WT* ox-LDL; **P = 0.004 *WT* ox-LDL *vs GSDME*[-/-], ox-LDL). **g** Western blotting analysis of *WT* and *GSDME*[-/-] peritoneal macrophages treated with ox-LDL. The experiments were repeated three times and the results were similar. **h** *WT* and *GSDME*[-/-] peritoneal macrophages were treated with TNF (100 ng/ml) +cycloheximide (CHX; 20 µg/ml) for the indicated time. LDH activity in the medium was measured. The fraction of cells was normalized by the experimental LDH release as a percentage of the positive controls. *n* = 8/group. ***P < 0.001. Unpaired two-tailed Student's *t* test was used for between-group comparisons (**d**, **e**). One-way ANOVA with Bonferroni post hoc test for multiple comparisons (**f**). Repetitive Measure Analysis of Variance ANOVA was used in analyzing LDH at different time **h**. For all panels, data are expressed as mean ± SEM. Each experiment was repeated independently three times for **d**–**h**. Source data are provided as a Source Data file.

phosphorylation inhibitor, protects against high-fat-induced atherosclerosis in *ApoE*[-/-] mice and rabbit[34,35]. Our previous data imply the significance of GSDME in the inflammatory response. Here, we hypothesized that STAT3 may be involved in the transcription regulation of *GSDME*. As shown in Fig. 3c, pSTAT3 (phosphorylated STAT3) levels were significantly increased in aortas derived from *ApoE*[-/-] mice fed on a western diet for 12 wk compared with those fed on a chow diet. Like *GSDME*, *STAT3* was also expressed in human atherosclerotic macrophages (Fig. 7a). Moreover, GSDME and pSTAT3 protein levels were simultaneously elevated in peritoneal macrophages treated with oxLDL or caspase 3 activator TNF (Fig. 7b, c). Based on these findings, we hypothesized that STAT3 might be involved in *GSDME* transcriptional regulation.

In overexpression or knockdown experiments, we found a direct correlation of STAT3 with GSDME expression (Fig. 7d–g). Increased GSDME expression was found in macrophages transfected with plasmid encoding *STAT3C* (a constitutively active version of *STAT3*)[36]. Consistently, knockdown of endogenous *STAT3* with small interfering RNA (siRNA) reduced GSDME expression. Bioinformatics also predicted the presence of 4 STAT3 binding sites in the promoter region of the *GSDME* gene (Fig. 7h). Chromatin immunoprecipitation assay demonstrated an increase in the binding of STAT3 to the 4 binding sites containing a consensus sequence TTCTGAGAAG (Fig. 7i). Furthermore, we observed that STAT3C increased *GSDME* promotor-driven luciferase activity (Fig. 7j). Collectively, these data indicate the significance of STAT3 in *GSDME* transcriptional regulation, implying a possible therapeutic approach for atherosclerosis.

## Discussion

Atherosclerosis is a chronic inflammatory disease characterized by macrophage death in lesion's lipid core[37]. In the past decades, putative therapeutic strategies targeting chronic inflammation in atherosclerosis have been proposed and may open a way for atherosclerosis protection[38]. Pyroptosis is inherently inflammatory and featured with rapid plasma-membrane rupture and proinflammatory cytokines release[39]. Previous studies have demonstrated macrophage inflammatory caspases and pyroptosis render lesion instability[40]. Thus, the molecules involved in macrophage pyroptosis might provide targets in atherosclerosis therapy. Meanwhile, atherosclerotic lesions are characterized by cellular heterogeneity and complexity. High-resolution technologies, including scRNA-seq (single-cell RNA sequencing), allow for a detailed analysis of the molecules and immune cells. Standard scRNA-seq data processing excluded mitochondrial gene content above 10%, indicating potential plasma membrane rupture and dissociation damage. In our study, we also considered that the mitochondrial gene content varies among different sample types which correlated with tissue type and pathological status[41]. *GSDME*

expression was found to be significantly positively correlated with mitochondrial gene content (Supplementary Fig. 2c). Due to high mitochondrial activity in atherosclerosis[42], we also included a chose a higher threshold of 50% to keep GSDME expression cells and remove dead and dying cells with reference to several other papers[25,26,43] and obtained consistent results with the threshold of 10%, indicating the robustness of our study. In conclusion, the scRNA-seq analyzes provide a high-resolution characterization of diverse cellular clusters and *GSDME* distributions in advanced human atherosclerotic plaques.

The field of pyroptosis has been greatly developed with the discovery of GSDM family. Among the family members, GSDMD is best-studied. Typically, GSDMD can be cleaved by inflammatory protease caspase-1 after canonical inflammasome activation with subsequent pyroptotic cell death and cytokine release[16]. GSDMD is involved in many pathophysiologic processes, including septic shock, spinal cord injury, and myocardial ischemia/reperfusion (I/R) injury[44–46]. *GSDMD* ablation blocks hypoxia/reoxygenation-induced cardiomyocyte pyroptosis and IL-18 release. Aberrant NLRP3 activation by oxidized LDL activates GSDMD and exacerbates atherosclerosis in mice and humans[47]. Recent studies reported that GSDMD mediates inflammation-induced defects in reverse cholesterol transport and promotes atherosclerosis. *GSDMD*[-/-] mice exhibit decreased atherosclerotic lesion area[48,49]. However, *GSDMD*-deficient cells are still susceptible to inflammasome-mediated cell death. Inflammasome stimulation of *GSDMD*-deficient cells leads to apoptotic caspase cleavage of GSDME.

GSDME, the second best-studied GSDM family member, was reported as a gene associated with non-syndromic hearing loss in humans[18]. All known mutations in *GSDME* gene result in transcriptional skipping of exon 8, thereby producing a truncated protein with cytotoxic activity[50,51]. The physiological activation mechanisms of GSDME appear to involve GSDME cleavage by caspase 3 and the formation of pores in the plasma membrane due to N-GSDME fragment oligomerization, triggering lytic cell death. N-GSDME also permeabilizes the mitochondrial membrane to augment caspase 3 activation during apoptosis and inflammasome activation[52]. Thus, GSDME has the ability to switch apoptosis into pyroptosis in response to caspase 3 activators such as chemotherapeutic drugs, tumor necrosis factor, and viral infection[20]. Previous studies have reported that *GSDME*[-/-] mice are protected from chemotherapy-induced tissue damage and weight loss[21] and proposed that caspase 3/GSDME pathway is a switch between apoptosis and pyroptosis in cancer[53]. Mai et al. reported that caspase 3-mediated GSDME activation contributes to cisplatin- and doxorubicin-induced secondary necrosis in mouse macrophages[54]. Liu et al also found that CAR-T cell therapy induces cytokine release syndrome (CRS) through GSDME-mediated pyroptosis in vivo and *GSDME* knockout eliminates CRS

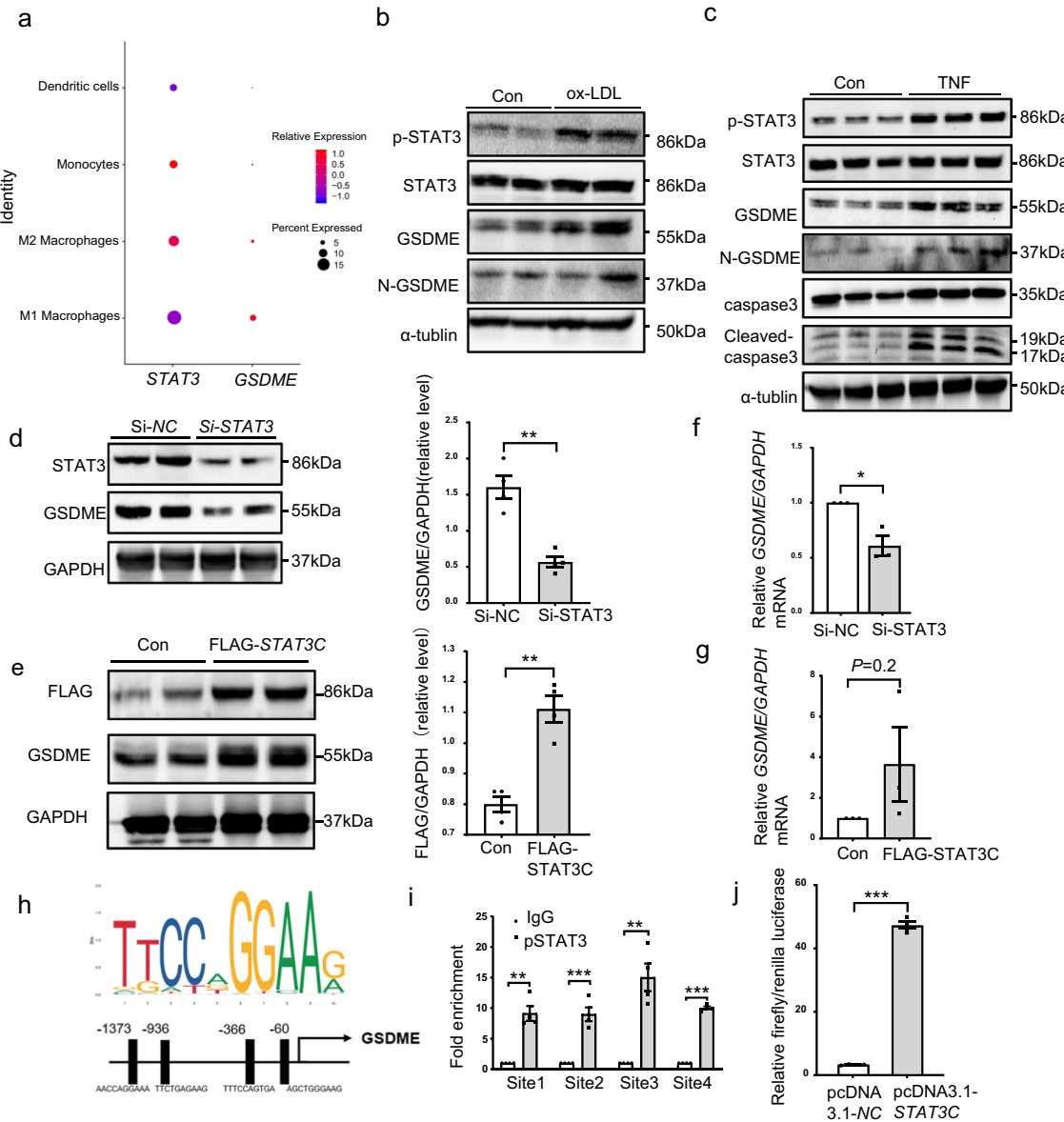

**Fig. 7 | STAT3 targets the *GSDME* promoter and activated *GSDME* transcription.**
**a** Dot plots showing the expression pattern of *STAT3* and *GSDME* genes among the
myeloid cell subgroups in human atherosclerotic plaques. Color scale represents
relative expression levels; blue: low, red: high. **b** Representative western blotting
of p-STAT3, STAT3, GSDME, and N-GSDME in mouse PMs treated with ox-LDL
(100 μg/ml) for 24 hours. **c** Western blotting analysis of p-STAT3, STAT3, GSDME,
N-GSDME, cleaved caspase 3 and caspase 3 in PMs treated with TNF (100 ng/ml)
for 24 hours. **d** PMs were transfected with siRNA control (Si-*NC*;100pmol) or
siRNA-*STAT3*(Si-*STAT3*;100pmol) for 72 hours, and GSDME protein were deter-
mined by western blotting (*n* = 4/group, **P = 0.001). **e** PMs were transfected with
FLAG-pcDNA3.1-*STAT3*(5 μg/ul) or pcDNA3.1-NC(5 μg/ul) for 48 hours, and
GSDME protein were determined by western blotting (*n* = 4/group, **P = 0.001).
**f** PMs were transfected with siRNA control (Si-*NC*;100pmol) or siRNA-*STAT3*(Si-
STAT3;100pmol) for 72 hours, and *GSDME* mRNA was determined by quantitative
polymerase reaction (*n* = 3/group, *P = 0.012). **g** PMs were transfected with FLAG-

pcDNA3.1-*STAT3*(5 μg/ul) or pcDNA3.1-NC(5 μg/ul) for 48 hours, and *GSDME*
mRNA was determined by quantitative polymerase reaction (*n* = 3/group,
*P* = 0.220). **h** Depiction of 4 putative stat3 binding sites, at -1373/-1364(site1),
-936/-927(site2), -366/-357(site3), and -60/-51(site4) bp upstream of transcription
initiation site in the mouse *GSDME* promoter. **i** Chromatin immunoprecipitation
analysis with antibodies against pSTAT3 or IgG, soluble chromatin from PMs, and
primers targeting the region spanning the 4 binding sites in the *GSDME* promoter
(*n* = 4/group, **P = 0.007 site1, **P = 0.008 site3, ***P < 0.001). **j** 293 T cells were
transfected with a mouse *GSDME* promoter-driven luciferase vector (0.4 μg) and
pcDNA3.1-*STAT3C* or pcDNA3.1-NC(0.2 μg). After 48 hours, the luciferase activity
was measured and normalized to Renilla activity (*n* = 4/group, ***P < 0.001). For
all panels, data are expressed as mean ± SEM. Unpaired two-tailed Student's *t* test
was used for between-group comparisons. Each experiment was repeated inde-
pendently three times. Source data are provided as a Source Data file.

occurrence in mouse models[55]. Endogenous GSDME activation per-
mits continuous IL-1β release and membrane leakage, even in
GSDMD-sufficient cells. Wang et al. proposed that NLRP3 activation
can induce inflammation and delay necrotic cell death in the absence
of *GSDMD* seemingly via GSDME-mediated pyroptosis[31,56,57]. Caspase
3/GSDME axis after NLRP3 activation has been identified as "a salvage
inflammatory pathway" when the canonical NLRP3-GSDMD signaling

is blocked. GSDME may serve as a potential therapeutic target for
treating inflammation-related diseases, such as atherosclerosis.
However, the mechanism of GSDME involvement in atherosclerosis
remains incompletely elucidated.

The present study has indicated the prominence of caspase 3 and
GSDME in the process of atherosclerotic macrophage pyroptosis. The
single-cell RNA sequencing verified the inflammatory essence of

human atherosclerosis and demonstrated the distribution of *GSDME* among the different subgroups in the total atherosclerotic cells. The results showed that *GSDME* is prominently expressed in myeloid cells along with some fibroblasts. Among the myeloid cells, M1 macrophages are the main source of *GSDME* expression suggesting *GSDME* is important in the inflammatory progression (Fig. 2b). The immunofluorescence colocalization of GSDME with human macrophage markers further validated our findings. Macrophages in atherosclerosis are highly heterogeneous and possess a broad range of surface markers, which can be used as " personalized signatures"[58]. CD68, a pan-macrophage marker shared by macrophages[59], showed reasonable colocalization with GSDME in atherosclerotic plaques. To exclude potential contribution of CD68[+] smooth muscle cells, more specific macrophage markers should be considered. Panels of surface antigens including CD14 (a receptor for bacterial lipopolysaccharide (LPS)) and CD16 (Fcγ receptor III) are found to be expressed on the surface of human monocytes and characterize monocyte heterogeneity[60,61]. Monocytes receiving stimuli enter the inflamed tissues and differentiate into macrophages accompanied by function alterations. Thus, these markers are less representative for macrophages in local inflammation tissues. CD163, a receptor for hemoglobin-haptoglobin complexes, is implicated in heme catabolism after intraplaque hemorrhage and demonstrated greater specificity as a marker of macrophages in paraffin-embedded tissue samples compared with CD68[62–64]. Therefore, we also assessed the colocalization of CD163 with GSDME in human advanced atherosclerotic plaques and concluded that GSDME is mainly expressed in macrophages (Supplementary Fig. 1f). Different from conventional immunofluorescence, multiplex immunofluorescence allows the simultaneous detection of multiple markers on a single tissue section utilizing tyrosine amplification system with the detection signal geometrically magnified[65], which explains the variations between Supplementary Fig. 1f and Fig. 2d. The results of sc-RNA sequencing showed that *GSDME* is mainly expressed in macrophages, although it can also be detected in other cell types such as fibroblasts and dendritic cells. Besides, not all the macrophages in atheroma express *GSDME*. The pseudo-time analysis showed that *CD68* and *GSDME* expression levels were simultaneously increased (Supplementary Fig. 2d), and *CD68* is strikingly enhanced by lipid stimulation[62]. In vitro, we also found that ox-LDL promotes GSDME expression in macrophages (Fig. 5a), and the activation of STAT3 is involved in the transcriptional regulation of *GSDME* (Fig. 7h–j). STAT3 is activated by rapid and transient tyrosine and serine phosphorylation. Upon STAT3 activation, the suppressors of cytokine signaling proteins are rapidly induced to suppress the cytokine signals[66]. Thus, some of the macrophages inside the atherosclerotic plaque may not be fully activated due to insufficient activation of STAT3 signaling, possibly providing mechanical explanations for the absence of GSDME in cells with positive macrophage markers. The expression mode can be heterogeneous and dynamic in macrophages during atherosclerosis. Future work on the heterogeneity and dynamics of macrophages will be needed to fully elucidate the relationship between different signatures in macrophages and GSDME expression.

We also directly confirmed the interaction between caspase 3 and GSDME in human atherosclerotic plaques by using in situ proximity ligation assay. Meanwhile, *GSDME* deficiency attenuates the progression of atherosclerosis in *ApoE*[−/−] mice fed with a high-fat diet. *GSDME* expression is upregulated in atherosclerotic lesions, and *GSDME* deficiency suppresses pyroptosis-related inflammation and decreases the development of atherosclerosis. Serum levels of inflammatory factors, including IL-1β, TNF, and MCP-1, are decreased in *GSDME*[−/−]/*ApoE*[−/−] compared to *ApoE*[−/−] mice. The protective effect of *GSDME* deficiency is mediated, at least in part, by inhibiting the local inflammatory response. *GSDME* ablation in vitro restrains ox-LDL-stimulated inflammation in macrophages and suppresses macrophage adhesion and migration. These results indicate that GSDME augments

caspase 3 activation and inflammation during the progression of atherosclerosis.

GSDME expression level determines the fate of tumor cells in response to the caspase 3 activators[53]. The caspase 3/GSDME signal pathway is a "switch" that can shift the balance between apoptosis and pyroptosis. At a high GSDME level, caspase 3 can cleave GSDME to trigger pyroptosis. Otherwise, it triggers apoptosis. However, the regulation mechanism of *GSDME* expression has not been completely understood. It has been previously reported that *GSDME* is regulated by p53, which is known to activate the transcription of numerous tumor suppressors[67]. Also, GSDME is potentially activated by phosphorylation[52]. Here, we identified the binding sites of STAT3 in the promoter of *GSDME* and verified that *GSDME* is a transcription target of STAT3. It has been reported that ox-LDL activates STAT3 transcription factor and promotes proinflammation in atherosclerotic plaques[32]. Aberrant STAT3 activation contributes to inflammation during atherosclerosis[33]. STAT3 is mainly activated by dimerization upon tyrosine and serine phosphorylation. The activated STAT3 dimers then translocated to the nucleus, where they bind to the consensus promoter sequences of the target genes to initiate transcription[68]. Consistently, our data demonstrate that ox-LDL activates STAT3 in parallel with increased *GSDME* expression, and *GSDME* transactivation partly explains the atherosclerotic and proinflammatory effects of STAT3. Collectively, the present study provides initial evidence that GSDME activation aggravates pyroptosis-related inflammation in atherosclerotic lesions and contributes to the development of atherosclerosis.

In conclusion, we have demonstrated that *GSDME* expression is upregulated during atherosclerosis, and *GSDME* deficiency reduces pyroptosis-related proinflammatory cytokine release in atherosclerotic plaques. As shown in Fig. 8, ox-LDL or other inflammatory cytokines such as TNF activates STAT3 by phosphorylation, and the activated STAT3 promotes the *GSDME* transcription. Then, the upregulated GSDME augments caspase 3 activity and promotes converting apoptosis to pyroptosis. These findings identify regulating *GSDME* transcription and caspase 3/GSDME pathway as a possible approach for reducing atherosclerosis.

## Methods
### Human sample collection
The study protocol was approved by the Research Ethics Committees of The First Affiliated Hospital of Xi'an Jiaotong University (Xi'an, Shaanxi, China; permit number: XJTU1AF-CRF-2017-006) and performed in accordance with the Declaration of Helsinki. Atherosclerotic plaques were obtained from 8 male and 2 female patients who underwent a carotid endarterectomy (CEA) procedure at The First Affiliated Hospital of Xi'an Jiao tong University. All patients were diagnosed with a transient ischemic attack (TIA) or cerebral infarction before surgery accompanied by clinical symptoms such as dizziness or cognization dysfunction. The average age is 60 years. 3 patients had diabetes. 5 patients had hypertension. 6 patients had a smoking history, and all the patients had a previous stroke. 4 patients underwent right carotid endarterectomy, and 6 patients underwent left carotid endarterectomy. The medical information of patients with carotid artery stenosis is shown in Supplementary Table 1. Informed consent was obtained from all participants. No compensation was provided to the participants in this study.

### Experimental animals
All animal procedures were approved by the Animal Care and Use Committees of Xi'an Jiaotong University (Xi'an, Shaanxi, China; permit number: 2018-81822005) for animal welfare. Experiments were performed according to the guidelines from the Directive 2010/63/EU of the European Parliament on animal protection. *ApoE*[−/−] (C57BL/6 J background) and *GSDME*[−/−]/*ApoE*[−/−] mice were purchased from

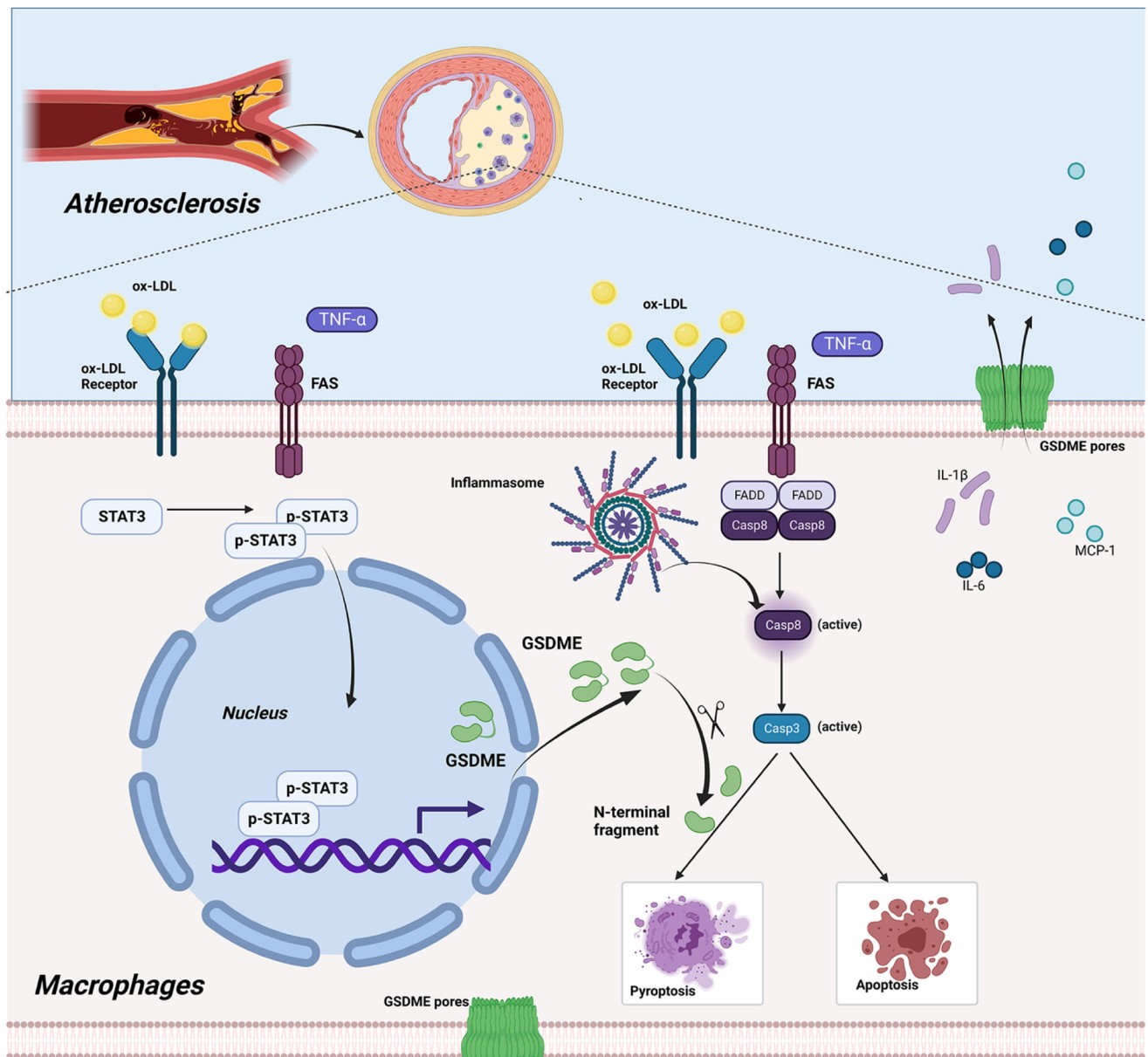

**Fig. 8 | Graphic summary of the mechanisms for GSDME accelerating inflammatory response in atherosclerosis.** Transcriptional activation and the upregulated *GSDME* augment the activity of caspase 3 and promote apoptosis converts to pyroptosis in macrophages during the process of atherosclerosis.

Shanghai Biomodel Organism. *GSDME*[−/−] mice used in the study were generated by co-microinjection of in vitro-translated Cas9 mRNA and gRNA into the C57BL/6 J zygotes to obtain F0 generation mice. Cas9 mRNA and gRNA were obtained by in vitro transcription. The gRNA sequence used to generate the knockout mice is ATGTGACCTGCTC GCTCTTCAGG. The genotypes were identified by PCR amplification and sequencing. *GSDME*[−/−]/*ApoE*[−/−] mice were generated by crossing *GSDME*[−/−] mice with *ApoE*[−/−] mice. Genotypes were determined by tail-snip PCR amplification. All mice were housed and bred at the Experiment Animal Center of Xi'an Jiaotong University under pathogen-free conditions. Animal procedures were performed in compliance with the guidelines from the Directive 2010/63/EU of the European Parliament on the protection of animals used for scientific purposes under a controlled environment ($20 \pm 2\,°C$, 12-h light/dark cycle) with free access to water and diet. We used only male mice in this study given that higher estrogen level in female mice is known to affect macrophage function and thus may complicate our data interpretation. A total number of 25 male 8 weeks aged *ApoE*[−/−] (C57BL/6 J background)

mice, 15 male 8 weeks aged *GSDME*[−/−]/*ApoE*[−/−] mice (C57BL/6 J background) and 10 male 8 weeks aged wild type mice (C57BL/6 J background) participated in our research. Mice were maintained on high-fat diet with 1.25% cholesterol and 40% (w/w) fat (D12108C; Research Diets) or a normal diet with 10% (w/w) fat (M10640 MolDiets) for 12 weeks. Mice intolerant to high-fat diet were excluded. Mice were euthanized by terminal inhalation of a 5% isoflurane/oxygen mixture until respiration came to a complete stop and reflexes in the animals' paws could no longer be triggered. After handling the mice, the investigators were blinded when assessing the outcome.

### Cell culture
For PMs, 1 mL of 3% thioglycolate medium was injected into the peritoneal cavity for 5 days. PMs were then isolated and cultured in DMEM high glucose medium supplemented with 10% fetal bovine serum, 100 units/mL penicillin, 100 μg/mL streptomycin, and 4 mmol/L L-glutamine in humidified air, 5% $CO_2$ at 37 °C. For BMDMs, bone marrow was isolated from femurs and tibias and

differentiated in DMEM high glucose medium containing 10% fetal bovine serum and 50 ng/mL with macrophage colony-stimulating factor (M-CSF; catalog number: 315-02-10, Pepro-tech) for 7 days.

## Proximity ligation assay

Interactions between caspase 3 and GSDME were detected by in situ proximity ligation assay (PLA) in human carotid atherosclerotic plaques. Following dewaxing and rehydration of tissue sections, antigen retrieval was performed by heating the slides for 30 min at 95°C in sodium citrate buffer and the PLA protocol was followed with Duolink In situ Brightfield kit according to the manufacturer's instructions (DUO 92012, Sigma), with incubation of the primary antibodies at 4 °C overnight. PLA minus and PLA plus probes (containing the secondary antibodies conjugated to oligonucleotides) were added and incubated for 2 h at 37 °C. More oligonucleotides were then added and allowed to hybridize into the PLA probes. Ligase was used to join the two hybridized oligonucleotides into a closed circle. The DNA was then amplified with rolling circle amplification, and detection of the amplicons was carried out using the Brightfield detection kit for chromogenic development.

## Immunohistochemistry

Deparaffinized tissue sections were treated with sodium citrate antigen retrieval buffer (catalog number: PR30001, Proteintech, Wuhan, China) Tissue sections were then incubated with 3% $H_2O_2$ for 10 min and 5% BSA blocking for 30 min at 37 °C. A primary antibody was applied, and the slides were incubated overnight at 4 °C. Signals were visualized using rabbit HRP-conjugated secondary antibody and SABC (catalog number: SA1028 Boster, Wuhan, China) and a hematoxylin (catalog number: MHS32, Sigma) counterstain. Image-Pro Plus 6.0 software (Media Cybernetics, USA) was used for analysis.

## Immunofluorescence staining

Human carotid atherosclerotic plaques were fixed in 4% paraformaldehyde. Then embedded in optimal cutting temperature compound (SAKURA O.C.T. Compound) and cut into 8-μm sections with a Leica cryostat. The frozen sections were dried at room temperature for 1 hour, followed by hydration with 1× PBS for 5 minutes. Tissue slides were treated with sodium citrate antigen retrieval buffer (PR30001, Proteintech, Wuhan, China), and then rinsed in PBS for 3× followed by blocking with 1% BSA and 0.25% Triton X100 in PBS for 1 hour at room temperature. Samples were then incubated with the primary antibodies against GSDME (Abcam, catalog number: ab230482, 1:100, Rabbit polyclonal) and CD68 (Proteintech, catalog number: 66231-2-Ig, 1:2000, Clone numbers: 3A9A7) at 4 °C overnight. After washing, the sections were incubated with a fluorescence-labeled secondary antibody for 30 minutes. The samples were then stained with DAPI for 5 minutes, followed by a 3× wash with PBS. Images were captured using an Olympus fluorescence microscope.

## Plasma lipid profile analysis

Mouse serum was collected after overnight fasting. Mouse plasma lipid profiles were measured by the automatic biochemical analyzer, type 7600 (Hitachi, Tokyo, Japan), in the department of clinical laboratory of The First Affiliated Hospital of Xi'an Jiao tong University.

## Atherosclerotic lesion analysis

After being fed a high-fat diet for 12 weeks, mice fasted for 14 h and were then anesthetized and euthanized. The heart and aortic tissues were removed from the ascending aorta to the ileal bifurcation and fixed in 4% paraformaldehyde. After fixation, the adventitia was thoroughly cleaned under a dissecting microscope. To analyze the lesion

area in the aortic root, the heart was dissected from the aorta, embedded in OCT compound, and sectioned (8-μm thickness). Six serial cryosections were collected from each mouse and stained with Oil Red O for neutral lipids, and then counterstained with hematoxylin to visualize nuclei. Images of plaques were captured under the Olympus microscope (BX51-FL-CCD), and quantitative analysis was performed with imageJ software (version 2.10) by averaging the lesion areas in the six sections. To analyze the lesion area in the aortic arch, the intimal surface was exposed by a longitudinal cut. Next, the aorta was stained for 2 hours in freshly prepared Oil Red O solution (0.3% Oil Red O in 60% isopropyl alcohol) at room temperature and destained several times with 70% ethanol. The stained aorta was then placed on a cationic anti-off slide and spread out completely. Images were captured using the high-resolution camera and analyzed using ImageJ software (version 2.10). Lesion areas were assessed as the percentage of Oil Red O positive area in the surface area of the entire aorta. Necrotic cores were defined as an area in which the extracellular matrix was lacking and replaced by dead cells or cellular debris[69].

## RNA-seq analysis

PMs were isolated from male $GSDME^{-/-}$ and $WT$ mice ($n = 3$) and then treated with ox-LDL (100 μg/mL) for 24 hours. After treatment, total RNAs were isolated from macrophages with Trizol reagent (catalog number: 15596026, Invitrogen, Carlsbad, CA, USA), and the sequencing libraries were prepared. Poly-A-containing mRNA was isolated from the total RNA by poly-T oligo-attached magnetic beads and then fragmented by an RNA fragmentation kit. The cDNA was synthesized using random primers through reverse transcription. After the ligation with the adaptor, the cDNA was amplified by 15 cycles of PCR, and then 200-bp fragments were isolated using gel electrophoresis. The products were sequenced by an Illumina HiSeq2500 instrument in Shanghai Majorbio Biopharm Technology (Shanghai, China). The data were analyzed on the free online platform of Majorbio Cloud Platform (www.majorbio.com). The differential expression of genes was selected by a fold change of >2 ($Log_2FC > 1$ or $Log_2FC < −1$) and $P$ adjust<0.05.

## Real-time quantitative polymerase chain reaction analysis

Total RNAs were isolated from tissues or cells using MiniBEST Universal RNA Extraction Kit (catalog number: 9767, Takara, Japan). cDNA was synthesized using PrimerScript RT Master Mix (catalog number: RR036A, Takara, Japan). A real-time polymerase chain reaction was performed using the FastStart Universal SYBR Green Master (catalog number: 06402712001, Roche, Mannheim, Germany). Individual quantitative RT-PCR was performed using gene-specific primers as shown in Supplementary Table 2.

## Western blotting analysis

Macrophages were harvested in ice-cold PBS and lysed in RIPA lysis buffer (catalog number: 06402712001, CST, Danvers, MA) in the presence of a protease inhibitor cocktail and phosphatase inhibitor cocktail (catalog number:11697498001, Roche). For protein analysis in the aorta, aortic tissues were homogenized and centrifuged at $14,000 \times g$ for 10 minutes at 4 °C. Proteins in the supernatant were quantified using the Pierce BCA Protein Assay Kit (catalog number: 23225, Thermo Fisher Scientific), separated by SDS-PAGE, and visualized by western blotting using specific antibodies. The relative intensities of the spots were determined by the Quantity One software (version 4.6.2) from Bio-Rad.

## Immunoprecipitation

Protein collection and quantification were performed sequentially as described above. The primary antibody was added to the cell lysate with gentle rocking overnight at 4 °C. Protein A or G agarose beads (catalog number:sc-2003, Santa Cruz Biotechnology) were subsequently added for 1 hour at 4 °C. Protein was obtained by centrifugation at $14,000 \times g$ for 10 min. The protein transfer, membrane

blocking, antibody incubation, and protein detection steps were similar to those performed for immunoblotting.

## TUNEL apoptosis assay
Mice aortic sinus sections were stained with the In Situ Cell Death Detection Kit (catalog number:11684817910, Roche) according to the manufacturer's recommendation.

## Lactate dehydrogenase (LDH) activity assay
LDH assay was performed with the Non-Radioactive Cytotoxicity Assay kit (catalog number: G1780) from Promega according to the manufacturer's recommendation.

## Binding site prediction
The potential STAT3 binding sites on selected mouse Gsdme genes were predicted by the use of the position weight matrix algorithm from JASPAR[70] to scan the promoter regions of the genes. The promoter regions were defined as −2000 to 500 from the transcriptional start site of the gene.

## Chromatin immunoprecipitation assay (ChIP)
ChIP assay was performed with the SimpleChIP Enzymatic Chromatin IP Kit (catalog number: 9003, CST), according to the manufacturer's recommendation. The sheared chromatin was precipitated with the STAT3 antibody (CST, catalog number: 12640, 1:100, Clone numbers: D3Z2G), or normal rabbit IgG (CST, catalog number: 2729, 5 µg for a single assay). Purified precipitated DNA was used as the template for qPCR and the primers used are listed in Supplementary Table 3.

## siRNA-mediated gene knockdown
Mice peritoneal macrophages were seeded into a plate and cultured to 80% confluence. siSTAT3 (sense5'3'GGGUCUCGGAAAUUUAACATT; antisense5'3'UGUUAAAUUUCCGAGACCCTT); SiNegativeControl (sense 5'3'UUCUUCGAACGUGUCACGUTT; antisense5'3'ACGUGACACGUUCG GAGAATT) were from (GenePharma, Shanghai, China). Cells were transfected with siRNA or negative control siRNA (30 nM/each siRNA) using Lipofectamine 3000 Reagent (catalog number: L3000001, Thermo Fisher Scientific) following the manufacturer's protocol.

## Plasmid construction and transfection
Desired DNA fragments of the mouse *GSDME* promoter were PCR-amplified and cloned into the pGL3-Basic luciferase reporter vector (Hanbio Biotechnology, Shanghai, China). All PCR products were validated by DNA sequencing. pcDNA3.1-*STAT3C* plasmid was purchased from Hanbio Biotechnology. 293 T cells were co-transfected with the luciferase and pcDNA3.1-*STAT3C* plasmids at 70–80% confluence, using X-tremeGENE HP DNA Transfection Reagent (catalog number: 06366236001 Roche). Forty-eight hours after infection, the *GSDME* promoter reporter activity was measured by firefly luciferase (catalog number: E1910, Promega) and normalized against control *Renilla* activity.

## In vitro scratch wound assay
Macrophages were incubated overnight yielding confluent monolayers for wounding. Wounds were made using a pipette tip and photographs were taken immediately (time zero) and 24 h after wounding, the distance migrated by the cell monolayer to close the wounded area during this period was measured. Results were expressed as a migration index.

## Transwell migration assay
Peritoneal macrophages isolated from WT mice or *GSDME*−/− mice were seeded onto transwell inserts with a polyethylene terephthalate membrane pore size of 8 µm (Thermo Fisher Scientific) in 24-well plates with the addition of MCP-1 (catalog number: 250-10, PeproTech) 600 ng/ml

to the medium in the lower chambers. After 24 h, media within the transwell inserts were carefully removed. Cells were fixed with 2% paraformaldehyde, permeabilized with 0.01% Triton X100 (catalog number: X100, Sigma-Aldrich), and stained with crystal violet (catalog number: C0775, Sigma-Aldrich). Cells that did not migrate across the transwell membrane were then removed by gently wiping with a cotton swab. Migrated cells were viewed with a phase-contrast microscope (Olympus BX51-FL-CCD) and processed using ImageJ software (version 2.10).

## Tissue dissociation and preparation for single-cell sequencing
GEXSCOPE Tissue Preservation Solution (Singleron) was used to store fresh atherosclerotic plaque. The specimens were washed twice with Hanks balanced salt solution and cut into 1–2 mm pieces. Tissue pieces were then digested for 15 min with 2 ml of GEXSCOPE Tissue dissociation solution (Singleron Biotechnologies, Nanjing, China) at 37 °C in a 15 ml centrifuge tube with uninterrupted agitation. After digestion, samples were filtered using 40-µm nylon sterile strainers and centrifuged at $350 \times g$ for 5 min. Then, the supernatant solution was removed and the sediment was resuspended in 1 ml of PBS (HyClone).

## Single-cell RNA sequencing (scRNA-seq)
Single-cell suspensions with $1 \times 10^5$ cells/ml in PBS (HyClone) were prepared. The suspensions were then loaded onto microfluidic devices and scRNA-seq libraries were constructed according to the Singleron GEXSCOPE protocol using the GEXSCOPE Single-Cell RNA Library Kit (Singleron,1110011)[71]. After quality checks, individual libraries were diluted to 4 nM and pooled for sequencing on an Illumina novaseq 6000 with 150 bp paired-end reads. After data normalizing, highly variable genes were identified and used for the following PCA (principal component analysis). Harmony v 0.1 be used to integrate samples and performed downstream analysis. Subsequently, clustering with 20 principal components and resolution 1.2 was performed by graph-based clustering and visualized using t-Distributed Stochastic Neighbor Embedding (t-SNE) or UMAP with Seurat functions RunTSNE and RunUMAP. Cells were filtered by gene counts between 200 to 5000 and UMI counts below 30,000. Cells with over 10% mitochondrial content were removed[25,26,72].

## scRNA-seq quantifications and statistical analysis
Sequencing outputs were demultiplexed to convert BCL files to FASTQ format using bcl2fastq (version 2.20). Briefly, sequencing data were processed using the CeleScope1.1.7 pipeline (Singleron https://github.com/singleron-RD/CeleScope/). Adapters and poly-A tails were trimmed (fastp V1) before aligning read two to GRCh38 using Ensemble v.92 gene annotation (fastp version 2.5.3a and featureCounts version 2.0.1). Reads with the same cell barcode, UMI, and gene were grouped to calculate the number of UMIs per gene per cell. The UMI count tables of each cellular barcode were used for further analysis. Cell type identification and clustering analysis were performed using the Seurat package (version 3.1.2). UMI count tables were loaded into R using the read. table function. The parameter resolution was set to 0.8 for the FindClusters function for clustering analyses. After STAR (version 2.6.1a) alignment and samtools (version 1.9) filtering[73,74], the filtered count matrix was separated by the cell type judgment result obtained by the CeleScope for further analysis. The cell type of each cluster was annotated by the known marker genes. The differentially expressed genes (DEGs) were generated by the function FindMarkers of Seurat with logFC. threshold 0.25 and min.pct 0.1; Genes with *P* value <0.05 in the differential genes list were selected and used for Gene ontology (GO) enrichment analysis by clusterProfiler (version 3.16.1);[75] For GSEA analysis, the differential genes were sorted by the avg logFC decreasingly. Based on the rearranged genes list, GSEA analysis was performed according to the GO database by the 'gseapy' python package

(version 0.9.15); Monocle (version 2.14.0) was used for trajectory analysis.

### Statistical analysis

Data were analyzed by IBM SPSS Statistics 23 software and presented as the means ± SEM. Data with normal distribution were analyzed using an unpaired two-tailed Student's $t$ test was used for between-group comparisons, and one-way ANOVA with Bonferroni post hoc test for multiple comparisons. Data with non-normal distribution were analyzed using the Mann–Whitney $U$ test to identify significant differences between two groups. Repetitive Measure Analysis of Variance (ANOVA) was used in analyzing LDH at different time points. All statistical tests were two-tailed, and $P < 0.05$ was considered statistically significant. The data plotting were performed with Graphpad Prism software 8.0.

### Reporting summary

Further information on research design is available in the Nature Portfolio Reporting Summary linked to this article.

### Data availability

The raw data of single-cell RNA-seq generated in this study have been deposited in Sequence Read Archive (SRA) database under accession code PRJNA802316. The raw bulk RNA sequencing data generated in this study have been deposited in Sequence Read Archive (SRA) database under accession code PRJNA802807; The publicly data used in this study are available in the GEO database under accession code GSE 43292. The remaining data generated in this study are provided in the Supplementary Information or Source Data file. Source data are provided with this paper.

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

## Acknowledgements

The authors thank professor Jianlin Liu and professor Shaoying Lu of the vascular surgeon teams of the First Affiliated Hospital of Xi'an Jiaotong University for the provision of human carotid plaques. The authors are grateful to Dr. Xiaolei Liu from Temple University for the help in revising the manuscript. This work was supported by the National Key R&D Program of China (2019YFA0802300 to Y.W., 2021YFA1301201 to Z.Y.Y.), the National Natural Science Foundation of China (81822005 to Y.W., 81970351 to Y.W., 81941005 and 92049203 to Z.Y.Y.), and the Clinical Research Award of the First Affiliated Hospital of Xi'an Jiaotong University, China (XJTU1AF-CRF-2017-006 to Y.W.). Figure 8 and Supplementary Fig. 1a (left) were created with BioRender.com.

## Author contributions

Y.Wu and Z.Y.Y. conceived and designed the research. Y.We, L.Y., and B.D.L. performed animal experiments. Y.We, X.X.Z., and L.L.C. performed all in vitro experiments. Y.We, L.Y., and T. Z. performed histologic analysis. Y.We and B.D.L. analyzed the data. G.T. helped revise the manuscript and draw the mechanism diagram. Y.Wu. and Z.Y.Y. supervised the work. Y.We wrote the original draft. All authors read and approved the final paper.

## Competing interests

The authors declare no competing interests.
