## [Peer Review File · Nature Communications]

GSDME-mediated Pyroptosis Promotes the Progression and Proinflammation of AtherosclerosisREVIEWER COMMENTS

Reviewer #1 (Remarks to the Author):

Wei et al. investigate the role of pyroptosis in atherosclerosis with a specific focus on its mediator GSDME. The authors provide data from cellular experiments, from human carotid endarterectomy samples and GSDME^{-/-}/ApoE^{-/-} mice. Single-cell transcriptome data shows that GSDME is mainly expressed in macrophages. Following this trail, the authors show with in vitro that ox-LDL induces the expression of GSDME and induces pyroptosis in macrophages. The authors also found a role of STAT3 in GSDME expression and postulate that they found a promising novel therapeutic approach for atherosclerosis.

Overall, the manuscript reports novel and interesting data. The presentation of the data and its presentation is good. The connection and relevance of some of the data is not clear. Also there are some mechanistic links missing, which the authors might be able to complete with some additional data.

The authors could include a recent summary of Qian et al. (Pyroptosis in the Initiation and Progression of Atherosclerosis. *Front. Pharmacol.* 2021;12:652963) in their introduction.

The authors describe the use of carotid plaques from 10 patients. Suppl. Table 1 describes some clinical characteristics. However, the indication for atherectomy is not included. Did all of these patients have strokes? Were the plaque considered to be unstable?

Figure 1:

The authors state: "Thus, although apoptosis clearly takes place in atherosclerotic plaques, pyroptosis appears to be a much more common mechanism for cell death."

However, the authors do not provide a quantification or overall supportive data for this statement.

The quantification of caspase 3 is missing.

Data in c does not look normally distributed. However, the authors describe the application of a t-test. N=3 seems to be low to make the important conclusions described.

Figure 2b and Suppl. Figure 2b: The expression profile of GSDME, especially in comparison to CD68 is not very impressive. Looking at this comparison, only a small number of monocytes/macrophages would express GSDME. These data are not consistent with the microscopy data shown in Figure 2d.

Figure 2d: A quantification would be helpful. Also the authors should provide a isotype control.

Data shown in Figure 2c, Suppl Figure 3 & 4 are interesting. However, these data seem not connected with GSDME? In fact the authors' statement "Taken together, these findings highlight cellular plasticity and suggest a role for macrophage GSDME in human advanced atherosclerosis." is only correct for the first part, which itself has no novelty. The authors would have to show GSDME changes along the plasticity described.

Figure 3: Would these data not be expected just based on the well known fact that mouse and human plaques contain macrophages, dependent on the stage of plaque development?

Figure 4: strong and central data. Instead of GSDME KO the authors should use GSDME ^{-/-} throughout this and other figures.

Figure 4: Can the authors provide further/better characterisation of the changes in atherosclerosis seen with the GSDME^{-/-} mice? Macrophage staining (MOMA-2), smooth muscle staining and other typical measures of atherosclerosis.

Pyroptosis of endothelial cells is proposed to be a central mechanism of early atherosclerosis (Qian et al. Pyroptosis in the Initiation and Progression of Atherosclerosis. Front. Pharmacol. 2021;12:652963). The authors data are not supportive of such a role of endothelial cells. Incubation of endothelia cells with ox-LDL and experiments such as shown in Figure 5 could clarify this important point.

The quality of Figur 6E should be improved.

Figure 6g does not seem suitable for a Student's t-test.

Figure 6: It is unclear how GSDME-deficiency actually influences migratory capabilities. Here convincing mechanistic data would clearly strengthen the manuscript.

Figure 8:

The role of the inflammasome should be listed.

The discussion is well written, however, overall a bit short.

The authors describe the targeting of GSDME as a potential therapeutic approach. Can the authors please investigate the capability of these mice to battle infections? are the GSDME-/- prone to infections? This is an important potential side effect of their proposed therapeutic approach.

Minor comments:

line 123 pyroptosis instead of proptosis.

Several blanks after full stops and commas are missing.

The authors should provide a list of abbreviations.

The authors state "The data underlying this article will be shared on reasonable request to the corresponding authors." For single-cell mRNA sequencing data it would be better to provide the raw data online.

The authors conclusions include the following sentence: "Then, the upregulated GSDME augments the activity of caspase3 and promote apoptosis converts to pyroptosis." Please improve the grammar and syntax here. In general, the paper would benefit from English editing.

Reviewer #2 (Remarks to the Author):

Comments:

This manuscript that investigates "GSDME-mediated Pyroptosis Promotes Atherosclerosis" presented a timely and novel in progression of atherosclerosis and transcriptional mechanism of GSDME. By using human samples and animal atherosclerosis model, this study have demonstrated that GSDME expression is upregulated during atherosclerosis and GSDME deficiency reduces pyroptosis related proinflammatory cytokine release in atherosclerotic plaques. It identified that caspase3 /GSDME pathway and the transcriptional regulation of GSDME as a novel approach for reducing atherosclerosis. It may represent a promising therapeutic approach for atherosclerosis. There are, however, several significant issues that will need to be addressed

Concerns that will need to be addressed:

1. Gasdermin family has six members, GSDMD is well studied, it has been demonstrated that

inflammasome activation triggers apoptotic cascade and GSDME cleavage in GSDMD-deficient macrophages (Zhou *et al*, Cell Reports April , 2021) and GSDMD promotes atherosclerosis (Opoku *et al*, Frontiers in Cell and Developmental Biology, 2021). In the current manuscripts the investigators investigated only the role of GSDME, what is the role of GSDMD in their model and responses- what is the role of redundancy? This has not been discussed or addressed at all. GSDMD needs to be discussed in this study.

2. Fig1c: the background is too much, cannot tell the GSDME, Casp3, and IL-1b positive staining. The atherosclerosis plaque picture and control vessels were on different magnification, the authors used "Protein IOD/Area" to compare the disease group and control was not accurate and should be addressed and fixed.

3. Supplemental Fig 1c, macrophages in lesion have no control staining (needs one), and more focused image is required to see the macrophages clearly.

4. Fig1d: Proximity ligation assay Casp3/GSDME, and IHC IL-1b, both staining have no controls.

5. Figure 3 showing the GSDME expression was increased in atherosclerosis. In Fig3a and Fig3 e, the authors compared GSDME protein and RNA expression level in early lesion and advanced lesion of patients, but in fig3 b and c, they compared GSDME protein and RNA *apoe*^{-/-} mice on normal diet and high fat diet. The panels in this figure jumped back and forth, it is better to reorganize the order of figure 3 for easier understanding.

6. The authors claimed that the estrogen level in female mice was known to affect macrophage function and thus only male mice were used in this study. But they collected patients' samples from both males and females, analyzed together. The authors should interpret the data separately.

7. Figure 4g showed IL-1b gene expression in whole aorta tissue, it is difficult to identify that if the IL-1b secretion was decreased in *Apoe*^{-/-}/*GSDME* compared to *Apoe*^{-/-} mice. Same concern for Figure 6f. These need to be addressed.

Reviewer #3 (Remarks to the Author):

The expression of GSDME in human and mouse plaque macrophages is interesting and novel, and is supported by histology and WB, if the specificity controls check out. The proposed involvement of GSDME in atherosclerosis through Casp8/Casp3 is also novel, but the data is not conclusive. The scRNA-Seq data is not convincing. There are serious problems with the mouse data (genetic background and *en face* lesions).

Major concerns:

1. Atherosclerosis is very sensitive to the genetic background. The *GSDME*^{-/-} mice were obtained from a commercial source. No information on genetic background of mice is given. If these mice were made in ES cells other than C57BL/6, for example 129 Sv, the authors must provide a genome-wide SNP map to determine the genetic background.

2. Figure 4D: the *en face* samples are of poor quality. On the left, much of the arch branches are preserved. On the right, most are missing, as are the mesenteric and renal arteries. This severely confounds the interpretation and the percent plaque area. Thus, the data is not valid.

3. Figure 2D, 3A, B: missing specificity controls.

4. What methods were used for batch effect correction, dead cell removal, doublet removal in scRNA-Seq?

5. Figure 2D: CD68 stains macrophages and secretory SMC. Some of the cells show a spindly shape and might be SMCs. A macrophage-specific marker must be used.

6. Figure 4E: serial sections are needed

7. Figure S1B: GSDME is found in only a handful of cells. GSDME is not shown in the sc heat map (S1A). Not convincing.

8. What are the macrophage numbers in plaques in WT vs GSDME KO mice?
9. Proinflammatory cytokine levels in WT vs KO mice (plasma) should be reported
10. The concentration of oxLDL used (100mg/ml) is very high and usually toxic as also reported in Figure 5f. To show that the oxLDL toxicity is mediated by GSDME through pyroptosis, Figure 5F experiment must be repeated also with GSDME KO macrophages.
11. GSDME expression should be analyzed also with different oxLDL doses.
12. The authors mostly report gene expression of IL1b and other proinflammatory cytokines in WT and GSDME KO mice after oxLDL or other stimuli. What about protein? The authors discuss about GSDME mediated pyroptosis in macrophages, protein levels of IL-1b and others in supernatants should be reported.
13. If GSDME, as proposed, promotes pyroptosis, how do you explain reduction in IL1b, TNF, and Mcp1 gene expression?
14. How does GSDME modulate the oxLDL signaling pathway? CASP8/CASP3 involvement in oxLDL mediated inflammatory pathway should be confirmed experimentally.

Minor comments

1. Figure 2C: symbols next to cell type too small to see color
2. Figure 3D: The symbols are not discernable.
3. Figure 6E: gene names are invisible
4. Full WB gels should be shown in supplementary material
5. Many language, grammar, word use issues

Point-by-point reply to the reviewers' comments

Reviewer #1

Wei et al. investigate the role of pyroptosis in atherosclerosis with a specific focus on its mediator GSDME. The authors provide data from cellular experiments, from human carotid endarterectomy samples and GSDME^{-/-}/ApoE^{-/-} mice. Single-cell transcriptome data shows that GSDME is mainly expressed in macrophages. Following this trail, the authors show with in vitro that ox-LDL induces the expression of GSDME and induces pyroptosis in macrophages. The authors also found a role of STAT3 in GSDME expression and postulate that they found a promising novel therapeutic approach for atherosclerosis. Overall, the manuscript reports novel and interesting data. The presentation of the data and its presentation is good. The connection and relevance of some of the data is not clear. Also, there are some mechanistic links missing, which the authors might be able to complete with some additional data.

Q1: *The authors could include a recent summary of Qian et al. (Pyroptosis in the Initiation and Progression of Atherosclerosis. Front. Pharmacol. 2021; 12:652963) in their introduction.*

Response: Thank you for your valuable suggestion. We have now cited this article in the introduction part of the revised manuscript (Introduction Page3 lines 53-57).

Q2: *The authors describe the use of carotid plaques from 10 patients. Suppl. Table 1 describes some clinical characteristics. However, the indication for atherectomy is not included. Did all of these patients have strokes? Were the plaques considered to be unstable?*

Response: Thank you for your expert comments. The patients mentioned in Suppl. Table 1 was admitted with complaints of dizziness and weakness of the upper limb. The criteria for subject selection were as follows: carotid stenosis of 50%-99% with stroke/Transient Ischemic Attack (TIA) symptoms in the past 6 months or carotid stenosis of 60-99% without stroke/TIA symptoms in the past 6 months (**Eur Heart J.** 2018, 39:763-816). In our study, all patients were diagnosed with Transient Ischemic Attack (TIA) or cerebral infarction before surgery. Moreover, we evaluated carotid artery stenosis through Doppler ultrasonography which indicated the thickness of the intima and the lumen stenosis exceeded 50%. The patients all met the surgical indications. We now added "All patients were diagnosed with Transient Ischemic Attack (TIA) or cerebral infarction before surgery accompanied with clinical symptoms such as dizziness or cognition dysfunction." at the page 16 line 334-335. The main complaints and the indication for atherectomy were added in Suppl. Table 1. The definition of unstable/vulnerable plaques was mainly based on morphological studies. Plaques displaying erosions, hemorrhages, or ulcers were identified as

unstable plaques. In contrast, plaques characterized by a smooth surface, no erosions, hemorrhages, or ulcers were considered stable (**American Heart Association**. 1995, 92:1355-1374).

The samples collected in our study included both stable and unstable plaques. Fresh carotid atherosclerotic plaques in our study were photographed and divided into stable plaques and unstable plaques according to their morphology. The representative pictures were shown in **Supplementary Fig. 1b**.

Q3: *The author's state: "Thus, although apoptosis takes place in atherosclerotic plaques, pyroptosis appears to be a much more common mechanism for cell death." However, the authors do not provide quantification or overall supportive data for this statement.*

Response: We thank the reviewer for the valuable point. We have now performed more precise characterization and analyses of cell death in atherosclerosis. Consistent with the previous study (**Atherosclerosis**. 1997,130:17-27), we found that the majority of cell death was characterized by membrane disruption and swelling by electron microscopy, and this data is now added in new **Supplementary Fig. 1c**.

Accordingly, we have revised the statement as "Thus, consistent with the previous study, although apoptosis takes place in atherosclerotic plaques, the majority of dying cells exhibit membrane disruption and cell lysis. (Supplementary Fig. 1c)" (Page 4, Line 82-83)

Q4: *The quantification of caspase 3 is missing*

Response: We have added the quantification of caspase 3 in **Fig. 1c**. The expression of caspase 3 was significantly increased in atherosclerotic plaques compared with the control vessels.

Q5: *Data in c does not look normally distributed. However, the authors describe the application of a t-test. N=3 seems to be below to make the important conclusions described.*

Response: We agree with the reviewer. The data was not normally distributed. Based on this, we have applied the Mann - Whitney U test and this test also reached significance ($p < 0.001$). Furthermore, to be more rigorous, we have analyzed more samples ($n=6$ for each group) and we have updated the detailed information in **Fig. 1c**.

Q6: *Figure 2b and Suppl. Figure 2b: The expression profile of GSDME, especially in comparison to CD68 is not very impressive. Looking at this comparison, only a small number of monocytes/macrophages would express GSDME. These data are not consistent with the microscopy data shown in Figure 2d.*

Response: We agree with the comment. This inconsistency was mainly because of the high requirement for cell viability in single-cell sequencing

experiments.

Obtaining fresh, high-quality biological samples and preparing highly active single-cell suspensions is the starting point for single-cell sequencing experiments. High cell viability and proper cell concentration are typical quality control requirements for single-cell processing and quality data. The total number of cells in the single-cell suspension prepared for use in our research is $> 1-2 \times 10^5$ and the percentage of the live cell is $> 85\%$.

GSDME expression underlays species differences in secondary necrotic death. Cells with high GSDME expression were more inclined to pyroptosis. Therefore, we humbly consider that GSDME-enriched cells are screened out during the preparation of single-cell suspensions because of the strict demand for cell viability.

Q7: *Figure 2d: A quantification would be helpful. Also, the authors should provide an isotype control*

Response: As suggested, we have added the results of the isotype control. Additionally, we have detected the expressions of CD68 and GSDME in normal vessels which were difficult to detect, and some non-specific staining of elastic fibers can be observed (**new Fig. 2d**).

The results are now described in the revised manuscript (Results, Page 5, Line 106-108).

Q8: *Data shown in Figure 2c, Suppl Figure 3 & 4 are interesting. However, these data seem not connected with GSDME? The authors' statement "Taken together, these findings highlight cellular plasticity and suggest a role for macrophage GSDME in human advanced atherosclerosis." is only correct for the first part, which itself has no novelty. The authors would have to show GSDME changes along the plasticity described.*

Response: Thanks for the suggestion. According to the result of immunofluorescence and single-cell analysis, GSDME is mainly expressed in atherosclerotic macrophages. Data are shown in Figure 2c modeled cell development during the progression of atherosclerosis which shows that macrophages are concentrated in the late stage of atherosclerosis. The result supports the hypothesis that GSDME mainly exerts its function at the late stage of atherosclerosis in macrophages.

According to the comment, we performed pseudotime developmental analysis of GSDME and macrophage marker CD68 to show the changes during the progression (**new Supplementary Fig. 2c**). The result showed that the expression level of CD68 and GSDME were all increased during the pseudo time which is consistent with those in the original manuscript.

Accordingly, the results are now described in the new manuscript (Results, Page 6, Line 116-119).

Q9: *Figure 3: Would these data not be expected just based on the well-known fact that mouse and human plaques contain macrophages, dependent on the stage of plaque development?*

Response: Thanks for the comment. Several studies have shown that GSDME-mediated pyroptosis participates in various diseases, such as acute kidney injury, obstructive nephropathy, doxorubicin-induced cardiotoxicity, and Cytokine release syndrome (CRS) (**Cell Death Differ** 2021, 28:2333-2350; **Life Sci** 2020, 242:117186; **Cell Death Dis** 2021,12:139; **Sci Immunol** 2020 5: eaax7969). However, the relationship between GSDME and atherosclerosis is poorly studied.

Indeed, it is well-known that mouse and human plaques contain macrophages. Analysis of single-cell sequencing in our study suggests that GSDME is predominantly expressed and functions in atherosclerotic macrophages, especially in advanced atherosclerosis. Our study is the first to show the significance of macrophage-associated GSDME during atherosclerotic progression. Accordingly, the data in Figure 3 strongly supported the important role of GSDME in the development of atherosclerosis in patient samples and mouse models. Nevertheless, we have re-organized Figure 3 so that it is more significant and relevant.

Q10: *Figure 4: strong and central data. Instead of GSDME KO, the authors should use GSDME -/-throughout this and other figures.*

Response: Thanks for the kind correction. We have replaced “GSDME KO” with “GSDME -/-” throughout the main text and figures.

Q11: *Figure4: Can the authors provide further/better characterization of the changes in atherosclerosis seen with the GSDME-/- mice? Macrophage staining (MOMA-2), smooth muscle staining, and other typical measures of atherosclerosis.*

Response: Thanks a lot for this constructive suggestion. As suggested, we performed immunostainings with antibodies against MOMA-2 and α -SMA to assess macrophage and smooth muscle cell contents. We also conducted Masson's trichrome staining to evaluate collagen content. The results have all been quantitatively analyzed and summarized in **Supplementary Fig. 6**. Collectively, these results indicate that the content of macrophages in the aorta roots was reduced in GSDME^{-/-}/ApoE^{-/-} mice. However, we did not observe smooth muscle cell changes (Page 8, Line 164-168).

Q12: *Pyroptosis of endothelial cells is proposed to be a central mechanism of early atherosclerosis (Qian et al. Pyroptosis in the Initiation and Progression of Atherosclerosis. Front. Pharmacol. 2021; 12:652963). The authors' data are not supportive of such a role of endothelial cells. Incubation of endothelial cells with ox-LDL and experiments such as shown in Figure 5 could clarify this important point*

Response: This is a valid concern. Endothelial cells are very important for atherosclerosis development. Previously, Shao Feng's data (*Nature* 2017, 547: 99-103) show HUVECs are GSDMEs-negative expressions, but we found there are GSDME-expressions in HUVECs. This important work by Prof. Shao Feng has been cited in the revised manuscript (Page 3 line 64).

Accordingly, we conducted the following experiment:

We isolated human umbilical vein endothelial cells (HUVECs) from human umbilical cords obtained from The First Affiliated Hospital of Xi'an Jiaotong University and performed a new experiment in which GSDME, caspase-3, p-STAT3, and STAT3 protein levels were analyzed by Western blotting in HUVEC treated with ox-LDL for 24h, and the representative results are shown as follows: GSDME was not sensitive to ox-LDL response in endothelial cells, in contrast, pstat3 and caspase 3 expression were significantly elevated in ox-LDL -treated endothelial cells, indicating the activation of pstat3 and caspase 3 in the procession is GSDME independent. Thus, we inferred that GSDME is not prominent in the ox-LDL-induced pyroptosis in endothelial cells.

Fig 1 in response to reviewer. Representative immunoblot for pSTAT3, STAT3, caspase 3, and GSDME. Data are mean±SEM from at least 3 independent experiments. The Student t-test was used. * p < 0.05.

Q13: The quality of Figure 6E should be improved.

Response: Thanks for your kind suggestions. The experiments were repeated, and the images have been replaced with those newly taken (**Fig. 6e**).

Q14: Figure 6g does not seem suitable for a Student's t-test.

Response: Thanks a lot for the kind reminder. We now have applied two-way repeated-measures ANOVA test to identify significant differences between the 2 groups. The results also reached significance ($p < 0.001$).

We have updated the "Statistical Analysis" section in the revised manuscript by adding the following sentence (Methods, Page 17, Line 361-362)

The figure legend was updated as well.

Q15: Figure 6: It is unclear how GSDME-deficiency influences migratory capabilities. Here convincing mechanistic data would strengthen the manuscript.

Response: Thanks for this insightful question. The relationship between Gasdermin and cell migration has been reported recently. Nitish Rana et al revealed that lack of GSDMB substantially impedes intestinal epithelial cells' wound closure by 57% (**Cell.** 2022,185:283-298), highlighting GSDMB-dependent epithelial cell migration. Prof. Lie Dai identified GSDME-mediated pyroptosis as a novel mechanism in regulating migration in Rheumatoid Arthritis Fibroblast-like Synoviocytes and reported that silencing GSDME suppressed RA-FLSs migration (**Front Cell Dev Biol.** 2021, 9: 810635).

Our GO data confirmed that GSDME ablation was related to the cell locomotion, biological adhesion, localization, and binding (GO:0040011; GO:0022610; GO:0051179; GO:0005488) which are essential in cell migration. Cell adhesion-related genes such as Lama2, Pcdhb, and Itga7 were significantly down-regulated in GSDME^{-/-} group and point to decreased migration in the absence of GSDME. Heatmap of cell locomotion, adhesion, and binding related genes in peritoneal macrophages induced by ox-LDL from WT and GSDME^{-/-} mice are shown as follows which indicates the mechanism of how GSDME-deficiency influences migratory capabilities. Further studies are needed on how GSDME affects the expression of these genes.

Fig 2 in response to reviewer. Heatmap of cell locomotion, adhesion, and binding related genes.

Another study has reported that MIF (macrophage migration inhibitory factor) release, which has chemotactic activity in monocytes, was highly upregulated in cells undergoing pyroptosis (**Immunol Cell Biol.** 2020, 98:782-790). Chemotaxis and migration are triggered by interactions between chemokines and their receptors. Thus, we humbly considered that the attenuation of inflammatory cytokine release in GSDME^{-/-} macrophages may involve in the

mechanism of GSDME-deficiency migratory capabilities.

Q16: *Figure 8: The role of the inflammasome should be listed.*

Response: Thanks for your kind advice. As suggested, we have added the inflammasome in Figure 8.

Q17: *The authors describe the targeting of GSDME as a potential therapeutic approach. Can the authors please investigate the capability of these mice to battle infections? are the GSDME^{-/-} prone to infections? This is an important potential side effect of their proposed therapeutic approach.*

Response: Intracellular lipopolysaccharide from Gram-negative bacteria including *Escherichia coli*, *Salmonella typhimurium*, *Shigella flexneri*, and *Burkholderia thailandensis* causes pyroptotic cell death, interleukin-1 β processing, and lethal septic shock (**Nature**. 2015, 526:666-671). To investigate if GSDME loss affects mouse mortality induced by LPS, WT and GSDME^{-/-} mice were challenged with lethal doses of LPS (25mg/kg intraperitoneally; n=12–13), and their mortality was monitored every 12h. Kaplan - Meier survival curves were used to analyze the data. The significance was evaluated by the log - rank (Mantel-Cox) test. There was no significant difference in survival rate between the two groups. Interestingly, a previous study (**Nature**. 2015, 526:666-671) has reported that Mice lacking *Gsdmd* were resistant to LPS-induced lethal septic shock and identified GSDMD as a key mediator of the host response against Gram-negative bacteria. Nevertheless, our data and the published work concluded that GSDME^{-/-} mice are not prone to infections in the setting of LPS challenge.

Fig 3 in response to reviewer. Kaplan–Meier survival curves for mice (n=12-13 for each genotype) challenged with 25 mg/kg LPS.

Thanks again for you giving us such excellent suggestions! We will further investigate the relationship between GSDME and infections in the future.

Minor comments:

Q1: line 123 pyroptosis instead of proptosis.

Response: Thanks for the kind reminder. We have corrected it.

Q2: Several blanks after full stops and commas are missing.

Response: Thanks for the reminder. We have corrected it.

Q3: The authors should provide a list of abbreviations.

Response: Thanks for your advice. The list of abbreviations has been added in the revised manuscript (Abbreviations, Page 18, Line 377-388).

Q4: The authors state “The data underlying this article will be shared on reasonable request to the corresponding authors.” For single-cell mRNA sequencing data it would be better to provide the raw data online.

Response: As suggested, the raw data of single-cell RNA-seq and transcriptomics generated in this study were deposited in Sequence Read Archive (SRA) with BioProject ID PRJNA802316; PRJNA802807; Submission-ID: SUB10997648; SUB11025050;

The SRA records will be accessible with the following link after the indicated release date (2023-02-14)

<https://www.ncbi.nlm.nih.gov/sra/PRJNA802316>

<https://www.ncbi.nlm.nih.gov/sra/PRJNA802807>

We are also glad to provide raw data before the indicated release date on a reasonable request.

Q5: The author’s conclusions include the following sentence: “Then, the upregulated GSDME augments the activity of caspase3 and promote apoptosis converts to pyroptosis.” Please improve the grammar and syntax here. In general, the paper would benefit from English editing.

Response: We apologize for our careless mistakes and appreciate your perusal of our manuscript.

The sentence has been re-written as follows:

“Then, the upregulated GSDME augments caspase 3 activity and promotes converting apoptosis to pyroptosis.”

The manuscript has been thoroughly revised and edited by a native speaker and tried to avoid grammar or syntax errors.

Reviewer #2

This manuscript that investigates “GSDME-mediated Pyroptosis Promotes Atherosclerosis” presented a timely and novel in progression of atherosclerosis and transcriptional mechanism of GSDME. By using human samples and animal atherosclerosis model, this study have demonstrated that GSDME expression is upregulated during atherosclerosis and GSDME deficiency reduces pyroptosis related proinflammatory cytokine release in atherosclerotic plaques. It identified that caspase3 /GSDME pathway and the transcriptional regulation of GSDME as a novel approach for reducing atherosclerosis. It may represent a promising therapeutic approach for atherosclerosis. There are, however, several significant issues that will need to be addressed

Q1: *Gasdermin family has six members, GSDMD is well studied, it has been demonstrated that inflammasome activation triggers apoptotic cascade and GSDME cleavage in GSDMD-deficient macrophages (Zhou, etc, Cell Reports April 2021) and GSDMD promotes atherosclerosis (Opoku, etc, Frontiers in Cell and Developmental Biology, 2021). In the current manuscripts the investigators invested only the role of GSDME, what is the role of GSDMD in their model and responses- what is the role of redundance? This has not been discussed or addressed at all. GSDMD needs to be discussed in this study.*

Response: Thanks for your kind advice. We have discussed this issue in detail in the Discussion Section in the revised main text (Discussion, Page 12, Line 253-264)

Q2: *Fig1c: the background is too much, cannot tell the GSDME, Casp3, and IL-1b positive staining. The atherosclerosis plaque picture and control vessels were on different magnification, the authors used “Protein IOD/Area” to compare the disease group and control was not accurate and should be addressed and fixed.*

Response: We apologized for the heavy background of the image. We then re-performed the immunohistochemical staining and replaced it with a **new Fig. 1c**. And the atherosclerosis plaque picture and control vessels were adjusted to the same magnification. We have changed the data, which are provided as a source data file.

Q3: *Supplemental Fig 1c, macrophages in lesion have no control staining (needs one), and a more focused image is required to see the macrophages.*

Response: This is a valid concern. We have added lesions in aortic sinus sections of male ApoE^{-/-} mice fed by normal diet as control and magnified the area with pyroptotic cell nuclei which are TUNEL positive as shown in **Supplementary Fig. 1e**.

Q4: *Fig1d: Proximity ligation assay Casp3/GSDME, and IHC IL-1b, both*

staining have no controls.

Response: As suggested, we have provided controls and present this picture in **new Fig. 2d**. Positive PLA signals were hardly detected in the control vessel.

Q5: Figure 3 showing the GSDME expression was increased in atherosclerosis. In Fig3a and Fig3 e, the authors compared GSDME protein and RNA expression levels in early lesion and advanced lesion of patients, but in fig3 b and c, they compared GSDME protein and RNA *ape-/-* mice on a normal diet and high fat diet. The panels in this figure jumped back and forth, it is better to reorganize the order of figure 3 for easier understanding.

Response: Thanks for your insightful advice. We agree that it is much better after reorganizing the order of the panels in figure 3. Based on the suggestion, we have reordered the panels. The results are now reordered in **new Fig. 3**.

Q6: The authors claimed that the estrogen level in female mice was known to affect macrophage function and thus only male mice were used in this study. But they collected patients' samples from both males and females, analyzed together. The authors should interpret the data separately

Response: Thanks for the comment. The patients were real-world consecutively collected including 8 men and 2 women. Both women were already in menopause and can exclude the effects of estrogen. We have added the detailed information in the Supplementary Table 1

Q7: Figure 4g showed IL-1b gene expression in whole aorta tissue, it is difficult to identify if the IL-1b secretion was decreased in *ApoE-/-/GSDME* compared to *ApoE-/-* mice. The same concern is for Figure 6f. These need to be addressed

Response: Thanks for this insightful question. Accordingly, we detected the inflammatory cytokine levels (IL-1 β , TNF- α , IL-6, and MCP-1) in the serum of *ApoE^{-/-}* and *GSDME^{-/-}/ApoE^{-/-}* mice (**Supplementary Fig. 7**). Because the amount of mouse serum is too limited to measure multiple inflammatory factors at the same time, we chose Luminex xMAP technology which could simultaneously measure many different analytes in a small sample volume (**Clinica Chimica Acta** 2006, 363:71-82; **Journal of immunological methods** 2000, 243: 243-255).

In correspondence with our in vitro data, the inflammatory cytokines including IL-1 β , TNF- α , and MCP-1 appeared to be decreased in *GSDME^{-/-}/ApoE^{-/-}* mice. These data are briefly discussed in the revised manuscript (Results, Page 8, Line 172-173).

Next, we performed a new experiment to detect the release of IL-1 β by WT and *GSDME^{-/-}* macrophages treated with ox-LDL. Proteins in the culture supernatant were concentrated through centrifugal filter tubes (Amicon Ultra-0.5), and protein in culture supernatant and cell lysates were subjected to Western blotting. The results are shown in **new Fig. 6g**. As expected, *GSDME^{-/-}* macrophages released mature IL-1 β to the cell-free supernatant more modestly

compared to WT macrophages indicating dependency on GSDME which is consistent with a previous study (**Cell Rep** 2021, 35:108998).

These data are briefly discussed in the revised manuscript (Results, Page10, Line 213-216).

Reviewer #3

The expression of GSDME in human and mouse plaque macrophages is interesting and novel, and is supported by histology and WB, if the specificity controls check out. The proposed involvement of GSDME in atherosclerosis through Casp8/Casp3 is also novel, but the data is not conclusive. The scRNA-Seq data is not convincing. There are serious problems with the mouse data (genetic background and en face lesions).

Q1: *Atherosclerosis is very sensitive to the genetic background. The GSDME^{-/-} mice were obtained from a commercial source. No information on the genetic background of mice is given. If these mice were made in ES cells other than C57BL/6, for example, 129 Sv, the authors must provide a genome-wide SNP map to determine the genetic background.*

Response: Thanks for your professional comment. The GSDME knockout mice used in our research were generated by co-microinjection of in vitro-translated Cas9 mRNA and gRNA into the C57BL/6J zygotes as described in the other study (**Nature** 2017, 547:99-103).

We also performed SNP loci detection to determine the genetic background, as well as the detailed information, is provided in the "Supplementary files". The genotyping of SNP loci demonstrated that the GSDME^{-/-} mice used in our research were the same as the C57BL/6J substrain by the SNP pattern (**Exp Anim** 2009, 58:141-149; **Genome Biol** 2013, 14: R82).

We have added the related description in the revised manuscript (Methods, Page 16, Line 340-341).

Q2: *Figure 4D: the en face samples are of poor quality. On the left, much of the arch branches are preserved. On the right, most are missing, as are the mesenteric and renal arteries. This severely confounds the interpretation and the percent plaque area. Thus, the data is not valid.*

Response: Thanks for pointing this out. We agree that the arch branches and mesenteric and renal arteries are confounding factors. Thus, lesion areas were assessed as the percentage of O.R.O positive area on the surface of the entire aorta excluding the branches. Furthermore, we tried to improve the quality of the en face samples. And the images have been replaced with **new Fig. 4d**. Atherosclerotic lesions were determined by using en face analysis of lesions on the intimal surface of the aorta⁴(Recommendation on Design, Execution, and Reporting of Animal Atherosclerosis Studies: A Scientific Statement from the American Heart Association), and the measurements were verified by a second researcher who was blinded to the experimental groups.

Q3: *Figure 2D, 3A, B: missing specificity controls.*

Response: According to this comment, we have added isotype controls and control vessels (human normal abdominal artery derived from autopsy) to the

results (**new Fig.2d**).

To make the results more organized, we have rearranged the order of Figure 3. Previous Figure 3A and Figure 3B are now in Figure 3B and Figure 3C.

In Figure 3B, we analyzed the protein expression of GSDME in human advanced carotid atherosclerotic plaques. Internal mammary artery (human healthy vessels) and early plaques were the controls. Additionally, we analyzed the expression of GSDME in different parts of the same plaque (**Supplementary Fig. 5**). The plaques were divided into three parts: Normal area, Area adjacent to the necrotic core, and Necrotic core. The normal area is the control. The result is described in the revised manuscript (Results, Page 7, Line 143-144)

In Figure 3C, we analyzed GSDME protein levels in atherosclerosis-prone mice model ApoE^{-/-} mice fed a high-fat diet or a normal laboratory diet. Aortas of ApoE^{-/-} mice fed on a high-fat diet (HFD) were used with a normal diet (ND) which grew fewer lesions as a control. In several other studies, ApoE^{-/-} mice fed on a normal diet were also used as control (**Arterioscler Thromb Vasc Biol** 2019, 39: 1787-1801; **Am J Transl Res** 2021, 13:1352-1364; **Life Sci** 2019, 232: 116590).

Q4: *What methods were used for batch effect correction, dead cell removal, and doublet removal in scRNA-Seq?*

Response: Thanks for the question. Methods used for batch effect correction, dead cell removal, and doublet removal were added in the Supplementary Information (Page 20, Line 281-288) as follows:

After data normalizing, highly variable genes were identified and used for the following Principal component analysis (PCA). Harmony v0.1 be used to integrate samples and performed downstream analysis. Subsequently, clustering with 20 principal components and resolution 1.2 was performed by graph-based clustering and visualized using t-Distributed Stochastic Neighbor Embedding (t-SNE) or Uniform Manifold Approximation and Projection (UMAP) with Seurat functions RunTSNE and RunUMAP.

Cells were filtered by gene counts between 200 to 5,000 and UMI counts below 30,000. Cells with over 50% mitochondrial content were removed.

Q5: *Figure 2D: CD68 stains macrophages and secretory SMC. Some of the cells show a spindly shape and might be SMCs. A macrophage-specific marker must be used.*

Response: This is a rigorous concern. To exclude a potential contribution of CD68 smooth muscle cells (SMC), we performed the double-labeling immunofluorescence assay and detected the expression of GSDME and a panel of other human macrophage markers (CD11b, CD14, and CD16) by fluorescence microscopy. We found that macrophages are the primary source of GSDME in human atherosclerotic plaques. And we have added the results in **Supplementary Fig. 1f**.

The results are described in the revised manuscript (Results, Page5, Line106-110)

Q6: *Figure 4E: serial sections are needed.*

Response: Thanks for the important suggestions. Based on this advice, representative sequential 12 sections (6 for HE;6 for O.R.O stain) in atherosclerosis-susceptible region of the aortic root (3 representatives for each group) were acquired in the study. The results were shown in **Supplementary Fig. 8**.

Q7: *Figure S1B: GSDME is found in only a handful of cells. GSDME is not shown in the sc heat map (S1A). Not convincing.*

Response: Thanks for this important comment. We agree with the reviewer's opinion that the expression profile of GSDME detected in the single-cell transcriptomic analysis is not very impressive. This was mainly because of the high requirement for cell viability in single-cell sequencing experiments. The total number of cells in the single-cell suspension prepared for use in our research is $> 1-2 \times 10^5$ and the percentage of the live cell is $> 85\%$. GSDME expression underlays species differences in secondary necrotic death (**Nat Commun** 2017, 8: 14128). Immunofluorescence detection revealed that GSDME was enriched in macrophages which were verified by different macrophage markers including CD68 (**new Fig. 2d and Supplementary Fig. 1e**). Therefore, we humbly consider that GSDME-enriched cells were screened out during the preparation of single-cell suspensions because of the strict demand for cell viability. Sc heat map (**Supplementary Fig. 2a**) only shows the top 10 marker genes per cluster, and the detected GSDME cannot be ranked in the top ten. Technical limitations have led to the need for further studies using multiple methods including single-cell sequencing.

Q8: *What are the macrophage numbers in plaques in WT vs GSDME KO mice?*

Response: Thanks for the comment. We conducted an immunofluorescence assay and detected the expression of MOMA-2 staining on the aorta root to reveal the macrophage composition.

As shown in **Supplementary Fig. 6**, the quantitative analysis of MOMA-2-stained sections indicated that the content of macrophage in the aorta roots was reduced in GSDME^{-/-}/ApoE^{-/-} mice, Relevant discussion has been added to the revised manuscript (Results, Page 8, Line 164-168).

Q9: *Proinflammatory cytokine levels in WT vs KO mice (plasma) should be reported*

Response: Thanks for the suggestion. To address this comment, we detected the inflammatory cytokine levels (IL-1 β , TNF- α , IL-6, and MCP-1) in the serum of ApoE^{-/-} and GSDME^{-/-}/ApoE^{-/-} mice (**Supplementary Fig. 7**). we chose Luminex xMAP technology which could simultaneously measure many different

analytes in a small sample volume because of the limited volume of mouse serum. As expected, the inflammatory cytokines including IL-1 β , TNF- α , and MCP-1 appeared to be decreased in GSDME^{-/-}ApoE^{-/-} mice. Reduced IL-6, although to a less extent, was also observed. These data are briefly discussed in the revised manuscript (Results, Page 8, Line 172-173).

Q10: *The concentration of oxLDL used (100mg/ml) is very high and usually toxic as also reported in Figure 5f. To show that the oxLDL toxicity is mediated by GSDME through pyroptosis, the Figure 5F experiment must be repeated also with GSDME KO macrophages.*

Response: Thanks for the insightful suggestion. Accordingly, we have conducted the experiment in which peritoneal macrophages from WT and GSDME^{-/-} mice were treated with ox-LDL in different concentrations for 24h. An LDH release assay was carried out to measure the release of cell content caused by pyroptosis. As shown in **Fig. 5f**, ox-LDL promotes cell death in a concentration-dependent manner. Meanwhile, ablation of GSDME attenuated LDH release indicating GSDME was required for ox-LDL induced pyroptosis. The results were briefly described in the revised manuscript (Results, Page 9, Line 194-195).

Q11: *GSDME expression should be analyzed also with different oxLDL doses.*

Response: Thanks for the constructive suggestion. To address the comment, we carried out western blotting to determine the expression of GSDME in macrophages treated with different concentrations of ox-LDL (0, 20 40, 80, or 100 ug/ml) for 24 h. The protein levels of GSDME were elevated as the concentration of ox-LDL increased (**Supplementary Fig. 9**). The results are now described in the revised manuscript (Results, Page 9, Line 180-183).

Q12: *The authors mostly report gene expression of IL1b and other proinflammatory cytokines in WT and GSDME KO mice after oxLDL or other stimuli. What about protein? The authors discuss GSDME mediated pyroptosis in macrophages, protein levels of IL-1b, and others in supernatants should be reported.*

Response: This is a valid and valuable concern. The inflammatory cytokine levels (IL-1 β , TNF- α , IL-6, and MCP-1) in the serum of ApoE^{-/-} and GSDME^{-/-}/ApoE^{-/-} mice which reflect systemic levels of inflammation were added in **Supplementary Fig. 7**. As expected, both local and systemic levels of inflammation were reduced in GSDME^{-/-}/ApoE^{-/-} mice. Prof. Bowen Zhou (**Cell Rep** 2021, 35: 108998) recently reported that GSDME is responsible for IL-1 β release after inflammasome activation. Thus, we focused on the expression of IL-1 β in vitro. We incubated peritoneal macrophages from WT and GSDME^{-/-} mice with ox-LDL, and protein in culture supernatant and cell lysates were subjected to Western blotting. As shown in **new Fig. 6g**, IL-1 β expression in GSDME^{-/-} macrophages was reduced and mature IL-1 β was released more

modestly.

The results are now described in the revised manuscript (Results, Page 8, Line 172-173) (Results, Page 10, Line 213-216).

Q13: *If GSDME, as proposed, promotes pyroptosis, how do you explain the reduction in IL1b, TNF, and Mcp1 gene expression?*

Response: Thanks for the comment. Previous studies have revealed that GSDME-mediated pyroptosis promotes inflammation in many pathophysiological processes including Dox-induced cardiac injury (**Life Sci** 2020, 242: 117186), cytokine release syndrome (**Sci Immunol** 2020, 5: eaax7969), obstructive nephropathy (**Cell Death Differ** 2021, 28: 2333-2350), and acute kidney injury (**Cell Death Dis** 2021, 12:139). Consistently, our study has revealed that GSDME deficiency attenuated atherosclerosis by inhibiting pyroptosis and inflammation. The inflammatory responses in the aorta from GSDME^{-/-}/ApoE^{-/-} mice were found alleviated compared with those from ApoE^{-/-} mice. A significant reduction in mRNA levels of proinflammatory genes such as TNF- α , IL-1 β , IL-6, and MCP-1 was observed in aorta from GSDME^{-/-}/ApoE^{-/-} mice. The findings from the current study provide new insights into the treatment of atherosclerosis.

Overall, Our and previous studies have indicated that GSDME-mediated pyroptosis and inflammation are closely related.

Q14: *How does GSDME modulate the oxLDL signaling pathway? CASP8/CASP3 involvement in oxLDL mediated inflammatory pathway should be confirmed experimentally.*

Response: Thanks for the comment. To examine the involvement of Casp3/Casp8 in ox-LDL-GSDME induced pyroptosis, we performed new experiments in which caspase-3, caspase-8, and GSDME were measured in macrophages treated with different concentrations of ox-LDL (0, 20 40, 80, or 100 ug/ml). Consistent with increased expression of GSDME, we observed that ox-LDL promoted a dose-dependent Casp3/Casp8 expression and activation in macrophages. Consistently, the expressions of NLRP3 are also in a dose-dependent manner in macrophages treated with ox-LDL. To further evaluate the role of NLRP3 in the ox-LDL-GSDME pathway, we treated macrophages with NLRP3-specific inhibitor MCC950. The results showed that the expression and activation of GSDME induced by ox-LDL were reduced in MCC950-treated macrophages. Overall, our results indicated that ox-LDL induces pyroptosis in macrophages through caspase 3/GSDME axis after NLRP3 activation. The relevant discussion has been added to the revised manuscript (**Results, Page 9 Line 180-187**).

Minor comments

Q1: *Figure 2C: symbols next to cell type too small to see color*

Response: Thanks for the advice. We have enlarged the symbols.

Q2: *Figure 3D: The symbols are not discernable.*

Response: We are sorry that we didn't make it clear. The symbols have been optimized.

Q3: *Figure 6E: gene names are invisible*

Response: Thanks very much. The resolution and scale of the images have been optimized accordingly. The original files have been supplied as "Cluster Heat Map Analysis Table" in Supplementary files to make it easier to see the details and gene names.

Q4: *Full WB gels should be shown in supplementary material*

Response: Full uncropped Western blots have been attached to the supplementary material.

Q5: *Many languages, grammar, word use issues*

Response: We apologize for our careless mistakes in this manuscript and the inconvenience caused during your reading. The manuscript has been thoroughly revised and edited by a native speaker. We hope it can meet the journal's standard. Thanks so much for your useful comments.

REVIEWER COMMENTS

Reviewer #1 (Remarks to the Author):

The authors have addressed my previous comments to my satisfaction and have now provided a stronger manuscript.

Reviewer #2 mediation by reviewer #1

I had a thorough look at the authors' response to reviewer 2.

The detailed feedback in regards to the individual reviewer questions are as follows:

Q1: The role of GSDMD is now well discussed in the revised manuscript.

Q2: The criticized pictures were exchanged. The new figure looks much better.

Q3: Requested controls have been added.

Q4: Again requested controls have been added.

Q5: Figure 3 is now well organised.

Q6: The issue of sex distribution seems to be addressed. However, I couldn't check Supplementary Table 1 as it was not available to me.

Q7: New data on inflammatory cytokines have been added.

Overall, I think the authors did a very good job to address the comments of reviewer 2.

Reviewer #3 (Remarks to the Author):

The authors edited their manuscript and included some new data.

1. New figure 4d: The authors took a pragmatic approach and removed all branches from the analysis. Although this is an unconventional approach, it is acceptable.

2. In the scRNA-Seq study, cells expressing up to 50% mitochondrial genes were retained. This is unacceptable. Mitochondrial genes in live cells are below 10%. Thus, the data presented contain many dead cells. One approach would be to color the UMAP by percent mitochondrial genes. This would show whether the dead cells cluster together. If the authors' hypothesis is correct, some of these dead and dying cells should express GSDME.

3. Figure S1F shows that CD14 and CD11b are mostly not colocalized with GSDME. Thus, classical monocytes, macrophages and neutrophils do not express GSDME. Only CD16 shows reasonable colocalization. CD16 is expressed by nonclassical monocytes and NK cells. The findings are misrepresented in the text.

4. Figure 5f (LDH release) is fraction of cells, not percent.

5. Figure S9: the α tubulin is overexposed.

Point-by-point reply to the reviewers' comments

First of all, we are most grateful for the reviewers' perusal of our manuscript and for the positive comment on the manuscript. And the point-by-point response is as follows:

Reviewer #1

The authors have addressed my previous comments to my satisfaction and have now provided a stronger manuscript.

Response: *Thanks for your previous great suggestions. It helps us a lot to improve our work.*

Reviewer #2 mediation by reviewer #1

I had a thorough look at the authors' response to reviewer 2. The detailed feedback in regards to the individual reviewer questions are as follows:

Q1: The role of GSDMD is now well discussed in the revised manuscript.

Q2: The criticized pictures were exchanged. The new figure looks much better.

Q3: Requested controls have been added.

Q4: Again requested controls have been added.

Q5: Figure 3 is now well organised.

Q6: The issue of sex distribution seems to be addressed. However, I couldn't check Supplementary Table 1 as it was not available to me.

Q7: New data on inflammatory cytokines have been added.

Overall, I think the authors did a very good job to address the comments of reviewer 2.

Response: *We would like to thank the reviewers for their comments and remarks on our earlier version of the manuscript to help to improve. Supplementary Table 1 was attached in supplementary information for further evaluation.*

Reviewer #3

1. New figure 4d: The authors took a pragmatic approach and removed all branches from the analysis. Although this is an unconventional approach, it is acceptable.

Response: *Thanks for your previous helpful suggestions. It helps us a lot to improve our work.*

2. In the scRNA-Seq study, cells expressing up to 50% mitochondrial genes were retained. This is unacceptable. Mitochondrial genes in live cells are below 10%. Thus, the data presented contain many dead cells. One approach would be to color the UMAP by percent mitochondrial genes. This would show whether the dead cells cluster together. If the authors' hypothesis is correct, some of these dead and dying cells should express GSDME.

Response: This is a valid and valuable concern. In our study, a more lenient criterion of 50% was chosen based on practical considerations. Nearly 1% of cells with high mitochondrial content were filtered out.

Although the criterion was 50%, the vast majority of cells had less than 10% of mitochondrial genes, and also the majority of M1 macrophages had less than 10% of mitochondrial genes (**Fig.1 in response letter**). In addition, we have divided total cells into 3 groups based on the expression of mitochondrial genes, and UMAP was colored by mitochondrial genes as well as GSDME. The number of cells with higher expression of mitochondrial genes (>40%) was too limited to be detected and the genes of these cells were largely degraded and thus can hardly be detected. In our study, we found that GSDME is predominantly expressed in macrophages, especially enriched in M1 macrophages in atherosclerosis. Therefore, we humbly consider that GSDME-enriched macrophages were screened out. Indeed, multiple tools and experimental approaches would be needed for a more nuanced evaluation.

Fig.1 in response letter. UMAP was colored by percent mitochondrial gene. UMAP was colored based on the interval of different expression contents of mitochondria.

3. Figure S1F shows that CD14 and CD11b are mostly not colocalized with GSDME. Thus, classical monocytes, macrophages, and neutrophils do not express GSDME. Only CD16 shows reasonable colocalization. CD16 is expressed by nonclassical monocytes and NK cells. The findings are misrepresented in the text.

Response: Thank you for the important comment. We re-assessed the colocalization of CD14 and CD11b with GSDME in different sites of the plaque. As shown in **Fig.2 in the response letter**, to some extent, GSDME and CD14/CD11b are co-localized, although not as well as CD16. Figure S1F has been updated.

CD163 is by far the most specific tissue cell marker and is expressed in monocytes of the peripheral circulation as well as in macrophages of most tissues. Compared with the CD68 antibodies, CD163 demonstrated greater specificity as a marker of monocyte /macrophage origin¹.

Thus, We next conducted multiplex immunofluorescence (mIF) that allowed simultaneous colocalization of three different human macrophage markers (CD68, CD163, CD16) with GSDME. All showed reasonable colocalization. And this data is now added in new **Supplementary Fig. 1g**. And the figure legend was also updated.

Accordingly, the results are now described in the new manuscript (Results, Page 5- 6, Line 108-113).

Fig.2 in response letter Immunofluorescence images of human macrophage markers (CD14 CD11b) and GSDME were co-staining in different sites of human carotid artery atheroma. Lu indicates lumen; and P, plaque. Scale bar: 100 μm ; Scale bar: 20 μm (magnification).

4. Figure 5f (LDH release) is fraction of cells, not percent.

Response: Thanks for the comment. we have updated the method for quantification according to the reviewer's comments.

5. Figure S9: the α tubulin is overexposed

Response: Thanks very much. We have corrected it.

References:

1. Lau SK, Chu PG, Weiss LM. CD163: a specific marker of macrophages in paraffin-embedded tissue samples. *American journal of clinical pathology* 122, 794-801 (2004).

Reviewers' comments:

Reviewer #3 (Remarks to the Author):

The new analysis shows that the scRNA-Seq data is of very low quality and therefore not valid.

All conjectured colocalizations do not exist or are not convincing.

Point-by-point reply to the reviewers' comments

Reviewer #3

Comment 1: The new analysis shows that the scRNA-Seq data is of very low quality and therefore not valid.

Response: For the sc-RNA results, the mitochondrial genes criteria we used in our study was controversial, and the quality of the sc-RNA seq data was doubted. Thus, We re-analyzed the data using new criteria that reviewer#3 proposed, and cells with <10% of the mitochondrial genes were further analyzed. The new results were consistent with our former conclusions that GSDME is mainly expressed in atherosclerotic macrophages which validates the original 50% mitochondrial cut-off. Accordingly, **Figure 2** and **Supplementary Figures 2-4** were replaced. Meanwhile, the analysis results based on 50% mitochondrial gene cut-off were toned down.

Indeed, it has been controversial to set the standard for mitochondrial transcript threshold in the field. The mitochondrial gene content differed between sample types which correlate with tissue type and pathological status (*Nat Commun.* 2018 Feb 13;9(1):490). Thus, we included a higher mitochondrial transcript threshold due to high mitochondrial activity in atherosclerosis (*Circ Res.* 2007 Mar 2;100(4):460-73.) Our study aimed to investigate GSDME-related pyroptosis in atherosclerosis, and some important genes will be ignored if the threshold is too low. Additional analysis also showed that GSDME expression was found to be significantly positively correlated with mitochondrial gene content (**Supplementary Figure.2c**). Therefore, we initially chose a threshold of 50% to optimize keeping GSDME-expression cells and removing dead and dying cells with reference to several other papers (*Cell.* 2021 Sep 2;184(18):4734-4752.e20 PMID: 34450029; *Nat Commun* 2022 Jan 10;13(1):181 PMID 35013299; *Nat Commun.* 2018 Oct 22;9(1):4383. PMID: 30348985). These papers are also cited in the revised manuscript (**Results, Page 6, Line 114-115**).

The number of cells that are analyzed for the scRNA analysis in both levels of the mitochondrial gene was added in the revised manuscript (**Results: Page5, line103; Page6, line 115**), and the figure legends (**Figure 2, and Supplementary Figure. 5**).

In addition, the reviewer also commented that "*if the authors' hypothesis is correct, some of the dead cells should express GSDME*". As suggested, we tried to analyze this part of the cells. In fact, dead or dying cells are hard to analyze for RNA degradation and most of the dying cells were screened out in sc-RNA data processing in our study. Thus, we analyzed the correlation between GSDME expression and mitochondrial gene content. As shown below, the expression of GSDME was positively correlated with the mitochondrial gene content which supports our proposed conclusions. The new result is added to the **new Supplementary Fig. 2c** and is now described in the revised

manuscript (Results, Page 6, Line 113-114).

new Supplementary Fig. 2c. GSDME is significantly positively correlated with mitochondrial gene content.

In conclusion, although the results indicate that the cell clusters at 50% cut-off are robust and consistent, the filter is not standard in the field. Thanks for the reviewer's insightful suggestions and we focused more on the results in the 10% cut-off. We have added a comment to this issue in the revised manuscript (Results, Page5 lines 100-104; Page5-6 lines 110-117) (Discussion, Page13 lines 268-280;).

Comment 2: All conjectured colocalizations do not exist or are not convincing.
Response: CD68 has been used repeatedly as a classical macrophage marker by several decent papers (*Circ Res.* 2016; 119:422-433 PMID:27256105; *JACC Basic to translational science.* 2018; 3:766-778 PMID:30623136; *Circ Res.* 2017; 121:1047-1057 PMID:28827412). Because previous studies suggest positive staining of CD68 in smooth muscle cells (*Circulation.* 2014 Apr 15;129(15):1551-9). it couldn't be excluded that the potential contribution of CD68⁺ smooth muscle cells may be involved in atherosclerosis. Thanks for the constructive suggestions from Reviewers, we've further validated other markers to determine their colocalization with GSDME in macrophages. Among them, CD163 demonstrated greater specificity as a marker of macrophages in paraffin-embedded tissue samples, compared with panels of macrophage surface antigens including CD68. CD163 is exclusively expressed in macrophages which are implicated in heme catabolism after intraplaque hemorrhage (*Immunol Rev.* 2014 Nov;262(1):36-55; *Immunogenetics.* 2001

Mar 53 (2):170-7).

Therefore, we conducted a new experimental method, 5-color multiplex immunofluorescence (mIF) that allowed simultaneous colocalization of three different human macrophage markers (CD68, CD163, CD16) with GSDME. To be more objective, the co-localization was quantified using the JaCoP plugin in image J (**Journal of microscopy. 2006; 224:213-232**). The Mander's coefficients for all three macrophage markers were over 95%. These results are sufficient enough to prove that GSDME is predominantly expressed in macrophages. Therefore, it is obviously unreasonable to conclude that "All conjectured colocalizations do not exist or are not convincing."

The results were shown as follows and updated in **Supplementary Fig.1f**. Furthermore, these results are also supported by our sc-RNA analysis (**Fig.2b, Supplementary Fig. 2d, Supplementary Fig. 5d**).

Different from conventional immunofluorescence, multiplex immunofluorescence (mIF) allows the simultaneous detection of multiple markers on a single tissue section (**Cancer Commun (Lond). 2020 Apr;40(4):135-153**). This method utilizes a tyrosine amplification system with the detection signal geometrically magnified which may potentially account for the variations between Supplementary Fig. 1f and Fig.2d.

GSDME is mainly expressed in macrophages. However, results of sc-RNA Sequencing showed that other cell types such as fibroblasts, and dendritic cells can also express GSDME, indicating a potential explanation for variations in co-localization results. In addition, The pseudo-time analysis showed that CD68 and GSDME expression levels were simultaneously increased (**Supplementary Fig. 2d**) and CD68 is strikingly enhanced by lipid stimulation. In vitro, we also found that ox-LDL promoted GSDME expression in macrophages (**Fig. 5a**) and the activation of STAT3 involved in the transcriptional regulation of GSDME (**Fig. 7h-j**). STAT3 is activated by rapid and transient tyrosine and serine phosphorylation. Once upon STAT3 activation, SOCS proteins are rapidly induced and suppress the cytokine signals (**Nat Rev Mol Cell Biol. 2002 Sep;3(9):651-62**). Thus, some of the macrophages inside the atherosclerotic plaque may not be fully activated due to insufficient activation of STAT3 signaling, possibly providing mechanical explanations for the absence of GSDME in cells with positive macrophage markers. The expression mode can be heterogeneous and dynamic in macrophages during atherosclerosis. Future work on the heterogeneity and dynamics of macrophages will be needed to fully elucidate the relationship between different signatures in macrophages and the GSDME expression.

We have added the discussion about the co-localization of GSDME and macrophage markers in the revised manuscript (**Discussion, page15-16, lines 315-350**).

Supplementary Fig. 1f. mIF images of human carotid artery atheroma sections labeled with DAPI (blue), GSDME (red), CD163 (green), CD16 (purple), and CD68 (yellow). Scale bar: 100 μ m. mIF, multiplex immunofluorescence.

REVIEWERS' COMMENTS

Reviewer #1 (Remarks to the Author):

The authors have addressed the remaining concerns to my satisfaction.

Reviewer #3 (Remarks to the Author):

adequately revised. The Manders coefficient makes the colocalizations more convincing, and the threshold of mt reads <10% is more reasonable

We would like to thank the reviewers for their careful review of our manuscript and their help to improve the interpretation of our findings. We are pleased to hear that the reviewers are satisfied with the additional work we did, and are happy to publish a suitably revised version of our manuscript.

Point-by-point reply to the reviewers' comments

Reviewer #1 (Remarks to the Author):

The authors have addressed the remaining concerns to my satisfaction.

Response: Thanks for your previous great suggestions. It helps us a lot to improve our work.

Reviewer #3 (Remarks to the Author): adequately revised. The Manders coefficient makes the colocalizations more convincing, and the threshold of mt reads <10% is more reasonable.

Response: we are grateful for your perusal of our manuscript and for your positive comment on our manuscript.